# Pharmacological targeting of MCL-1 promotes mitophagy and improves disease pathologies in an Alzheimer's disease mouse model

Xufeng Cen[1,6], Yanying Chen[1,6], Xiaoyan Xu[1,6], Ronghai Wu[1,6], Fusheng He[2,6], Qingwei Zhao[4], Qiming Sun [1], Cong Yi[1], Jie Wu [2✉], Ayaz Najafov [3✉] & Hongguang Xia [1,4,5✉]

There is increasing evidence that inducing neuronal mitophagy can be used as a therapeutic intervention for Alzheimer's disease. Here, we screen a library of 2024 FDA-approved drugs or drug candidates, revealing UMI-77 as an unexpected mitophagy activator. UMI-77 is an established BH3-mimetic for MCL-1 and was developed to induce apoptosis in cancer cells. We found that at sub-lethal doses, UMI-77 potently induces mitophagy, independent of apoptosis. Our mechanistic studies discovered that MCL-1 is a mitophagy receptor and directly binds to LC3A. Finally, we found that UMI-77 can induce mitophagy in vivo and that it effectively reverses molecular and behavioral phenotypes in the APP/PS1 mouse model of Alzheimer's disease. Our findings shed light on the mechanisms of mitophagy, reveal that MCL-1 is a mitophagy receptor that can be targeted to induce mitophagy, and identify MCL-1 as a drug target for therapeutic intervention in Alzheimer's disease.

[1] Department of Biochemistry & Molecular Medical Center, Zhejiang University School of Medicine, Hangzhou 310058, China. [2] School of Pharmaceutical and Materials Engineering & Institute for Advanced Studies, Taizhou University, 1139 Shifu Avenue, Taizhou 318000, China. [3] Department of Cell Biology, Harvard Medical School, Boston, MA 02115, USA. [4] Research Center for Clinical Pharmacy & Key Laboratory for Drug Evaluation and Clinical Research of Zhejiang Province, The First Affiliated Hospital, Zhejiang University School of Medicine, Hangzhou 310003, China. [5] Zhejiang Laboratory for Systems & Precision Medicine, Zhejiang University Medical Center, 1369 West Wenyi Road, Hangzhou 311121, China. [6] These authors contributed equally: Xufeng Cen, Yanying Chen, Xiaoyan Xu, Ronghai Wu, Fusheng He. ✉email: jie_wu@fudan.edu.cn; ayaz_najafov@hms.harvard.edu; hongguangxia@zju.edu.cn

Alzheimer's disease (AD) is a severe neurodegenerative disorder with memory loss and cognitive dysfunction as its main symptoms. Mitochondrial dysfunction is a fundamental pathological hallmark of AD, as damaged neuronal mitochondria have been found to accumulate both in sporadic and familial types of the disease, as well as in AD animal models[1,2]. The impaired mitochondrial function triggers energetic stress that promotes the disease-defining amyloid-β (Aβ) oligomers and hyper-phosphorylated Tau (pTau) pathologies[1,3]. Impairment of the mitochondrial biogenesis, for example, induced by decreased expression of the mitochondrial biogenesis regulator PGC-1α, contributes to synaptic dysfunction and neuronal degeneration in AD[4,5]. Furthermore, mitochondria regulate cytosolic calcium homeostasis, and mitochondrial dysfunction-induced intracellular calcium imbalance leads to neuronal death and is implicated in AD[6]. Importantly, mitochondrial dysfunction has been reported to accelerate Aβ production and occur prior to the accumulation of Aβ deposits in the brains of AD mouse models[2,7,8]. Moreover, suppression of mitochondrial function using toxins or genetic deletion of mitochondrial proteins exacerbates Aβ pathology[9,10]. In light of this cumulative evidence, mitochondrial dysfunction has been suggested as a pivotal event in the initiation of AD, and interventions that bolster mitochondrial health may ameliorate the neurodegenerative pathologies associated with it[1,11,12].

Mitophagy is a selective autophagy pathway for mitochondrial quality control, in which engulfment of damaged or depolarized mitochondria by a double-membrane autophagosomal structure is followed by fusion with lysosomes for degradation[13]. Emerging findings suggest that mitophagy is also compromised in AD, resulting in the accumulation of dysfunctional mitochondria that contributes to synaptic dysfunction and cognitive decline in AD[11,14]. Conversely, mitophagy enhancement reduces Aβ plaques and Tau tangles in human neuronal cells and ameliorates memory impairment in transgenic mouse models of AD[11].

Mitochondria are targeted to mitophagy via various mitophagy receptors, including the ubiquitin-binding receptors optineurin (OPTN), p62 (SQSTM1), NDP52, and NBR1, as well as NIX/BNIP3L, FUNDC1, AMBRA1, Bcl-2-L-13, and Prohibitin 2 (PHB2), which can mediate mitophagy in a ubiquitin-independent manner[15–17]. Mitophagy receptors bind to LC3 family proteins via a four-residue-long "LC3-interacting region" (LIR) motifs ([W/Y/F]XX[I/L/V]) to promote elongation and closure of the phagophore membranes, thereby engulfing the mitochondria[18,19].

Here, our small-molecule compound screen of 2024 FDA-approved drugs or drug candidates revealed a BH3-mimetic UMI-77 as a potent mitophagy promoter. We identified the target of UMI-77, a key anti-apoptotic protein MCL-1, as a mitophagy receptor that interacts with LC3A to promote mitophagy. Finally, we found that UMI-77-induced mitophagy significantly improves AD pathologies seen in the APP/PS1 mouse model.

## Results

**UMI-77 selectively induces mitophagy independent of mitochondrial damage and apoptosis.** mt-Keima is a fusion of a mitochondrial signal sequence from cytochrome C oxidase subunit IV with the fluorescent protein Keima. This chimeric protein has been previously established as a robust sensor of mitophagy[20,21]. mt-Keima undergoes an excitation wavelength shift at low pH, allowing quantification of the mitochondria that have been exposed to the acidic milieu of lysosomes and hence, mitophagy. To identify mitophagy activators, we performed a high-throughput screen using a HEK293T cell line stably expressing mt-Keima and a library of 2024 FDA-approved drugs or drug candidates. We identified 20

activators of mitophagy that increase relative mitophagy levels by >1.5-fold (Fig. 1a, Supplementary Fig. 1a). Interestingly, some of the Bcl-2 family inhibitors (also known as BH3-mimetics), such as UMI-77, were found to trigger mitophagy.

UMI-77 is an MCL-1-specific compound that blocks the interaction between MCL-1 and Bax/Bak, thereby allowing Bax/Bak to induce apoptosis[22]. Unlike CCCP (carbonyl cyanide 3-chlorophenylhydrazone), UMI-77 did not induce mitochondrial damage in HEK293T and HeLa cells at the sub-lethal 5 μM dose (Fig. 1b). Compared with CCCP, the UMI-77+CCCP combination treatment significantly augmented mitophagy levels (Fig. 1c). These results indicated that UMI-77 can promote mitophagy independent of mitochondrial damage and can augment mitophagy triggered by CCCP-induced mitochondrial damage.

To eliminate the possibility that apoptosis induction may non-specifically induce the excitation shift of mt-Keima, we assessed the effect of all apoptosis inducers in our drug library on this established mitophagy reporter. As shown in Supplementary Fig. 1b, most apoptosis-inducing drugs did not trigger the mt-Keima excitation shift. Moreover, mitophagy induction by UMI-77 could not be rescued by pan-caspase inhibitor Z-VAD-fmk, suggesting that this UMI-77 can induce mitophagy independent of its established role in activating apoptosis (Fig. 1d). In addition, UMI-77 strongly induced mitophagy, but could not induce apoptosis (Supplementary Figs. 1e and 2a–d). These results strongly indicated that the induction of mt-Keima excitation shift by UMI-77 is independent of apoptosis induction and that UMI-77 can induce mitophagy, but not apoptosis at sub-lethal doses.

Consistent with the pro-mitophagy effect of UMI-77, using live-cell imaging of HEK293T-mt-Keima cells, we found that this compound induces co-localization of mitochondria with lysosomes (Fig. 1e). Transmission electron microscopy (TEM) confirmed that mitochondria were accumulated in autophagosomes of HeLa cells (Fig. 1f) and HEK293T cells (Supplementary Fig. 1f) following UMI-77 treatment. We found that UMI-77 promoted degradation of mitochondrial proteins (outer-membrane protein Tom20 and inner-membrane protein Tim23), but not that of endoplasmic reticulum marker calnexin or cytosolic marker tubulin in HEK293T, HeLa, SH-SY5Y, and U2OS cells (Fig. 1g, Supplementary Fig. 1g). The UMI-77-induced degradation of mitochondrial proteins could be blocked by the lysosome inhibitors E64D or NH$_4$Cl/Leupeptin, but not the proteasomal inhibitor MG-132 (Fig. 1h, Supplementary Fig. 1h). As E64D was able to increase the levels of LC3-II, UMI-77 may increase the autophagic flux (Supplementary Fig. 1c).

Macroautophagy is known to degrade mitochondria in a non-selective manner; however, we found that p62 levels (macroautophagy marker) and other organelle markers did not decrease following UMI-77 treatment, consistent with the notion that this compound induces mitophagy, but not macroautophagy (Supplementary Fig. 1d, i). Taken together, our findings indicate that the MCL-1-targeting BH3-mimetic UMI-77 specifically induces mitochondrial degradation via mitophagy, independent of macroautophagy, mitochondrial damage, or apoptosis.

**MCL-1 promotes mitophagy.** As MCL-1 is the de facto target of UMI-77, we investigated whether this anti-apoptotic protein participates in the UMI-77-driven mitophagy activation. As shown in Fig. 2a, knockdown of MCL-1 rescued the UMI-77-induced degradation of mitochondrial proteins Tom20 and Tim23 in HEK293T and HeLa cells. In addition, the MCL-1 knockdown prevented the induction of the mt-Keima excitation shift induced by the UMI-77 treatment (Fig. 2b and Supplementary Fig. 3a). These results indicate that MCL-1 is required for UMI-77-induced mitophagy activation.

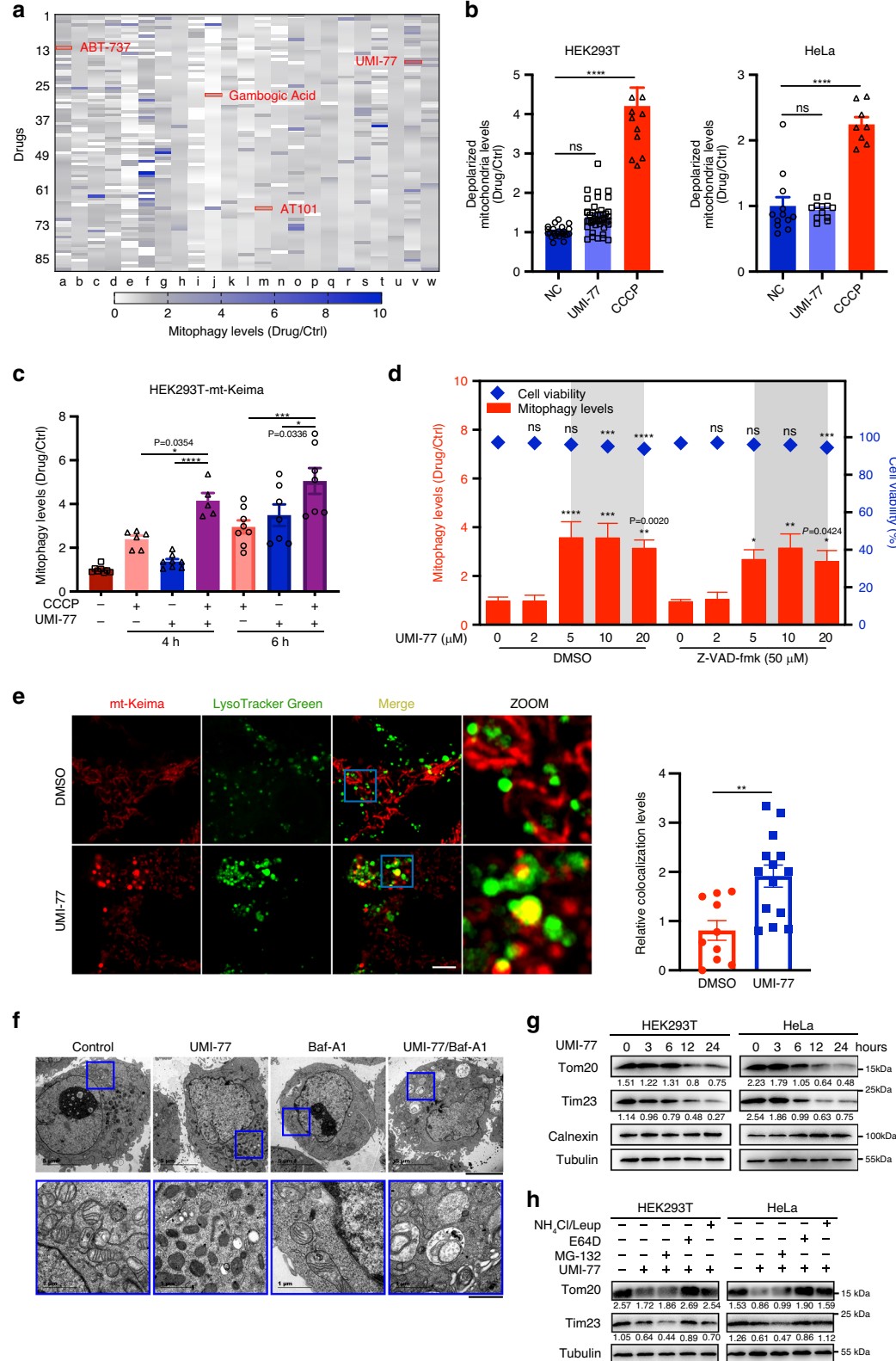

Overexpression of mitophagy receptors such as FUNDC1 and FKBP8 has been shown to promote mitophagy[23,24]. To investigate the role of MCL-1 in mitophagy, we generated a HEK293T stable cell line expressing MCL-1 in a doxycycline-inducible manner (HEK293T-MF2). Notably, doxycycline-induced MCL-1 overexpression resulted in the degradation of mitochondrial markers (Fig. 2d). Moreover, MCL-1

overexpression resulted in the co-localization of mitochondria with lysosomes (Fig. 2c). Interestingly, as shown in Fig. 2c, mitochondria became smaller and fragmented following MCL-1 overexpression, consistent with the notion of mitophagy induction. Finally, electron microscopy confirmed that MCL-1 overexpression drives mitophagy (Fig. 2e). These results are similar to the phenotype of Bcl-2-L-13 overexpression[25]. Taken

**Fig. 1 UMI-77 can induce mitophagy, without inducing mitochondrial damage or apoptosis. a** 2024 FDA-approved drugs or drug candidates were used to screen for mitophagy inducers. Each color block represents a drug in the screen. Bcl-2 family proteins inhibitors are indicated in red. **b** HEK293T and HeLa cells treated with UMI-77 (5 µM) or CCCP (10 µM) for 12 h. Mitochondrial membrane potential was assessed using JC1. One-way ANOVA (data represent mean ± S.E.M.; HEK293T: NC ($n = 24$), UMI-77 ($n = 45$), CCCP ($n = 12$); HeLa: NC ($n = 12$), UMI-77 ($n = 12$), CCCP ($n = 8$). ****$p < 0.0001$. ns, not significant). **c** HEK293T-mt-Keima cells were treated with UMI-77 (5 µM) combined with CCCP (10 µM) for 4 or 6 h. The mitophagy levels were analyzed by one-way ANOVA (data represent mean ± S.E.M. The sample size was, in turn, $n = 8$, $n = 6$, $n = 8$, $n = 5$, $n = 8$, $n = 7$, $n = 7$. ****$p < 0.0001$, ***$p < 0.001$, *$p < 0.05$). **d** HEK293T cells were transfected with pcDNA3.1-mt-Keima (mitophagy reporter) plasmid and treated with UMI-77 at indicated concentrations with or without Z-VAD-fmk (50 µM) for 12 h. Cell viability was determined using LIVE/DEAD™ imaging kit. One-way ANOVA (data represent mean ± S.E.M.; $n = 4$. ****$p < 0.0001$, ***$p < 0.001$, **$p < 0.01$, *$p < 0.05$. ns, not significant). **e** HEK293T-mt-Keima cells were stained with LysoTracker Green for 30 min, treated with 5 µM UMI-77. Scale bars, 5 µm. Quantification of the number of LysoTracker-positive dots colocalized with mt-Keima in cells were analyzed by two-tailed t test. (mean ± S.E.M.; DMSO ($n = 10$), UMI-77 ($n = 14$). **$p < 0.01$ ($P = 0.0021$)). **f** HeLa cells were treated with UMI-77 and Bafilomycin A1 (Baf-A1) for 8 h and analyzed by transmission electron microscopy. Scale bars, 5 µm; insets: Scale bar, 1 µm. **g** cells were treated with 5 µM UMI-77 for the indicated times and cell lysates were immunoblotted with indicated antibodies. The numbers under the blots represent the gray scale quantification (Tom20/Tubulin, Tim23/Tubulin). **h** cells were treated with 5 µM UMI-77 in the presence or absence of MG-132, E64D, and NH$_4$Cl/Leupeptin (Leup) for 12 h, and the mitochondrial marker proteins (Tom20, Tim23) were detected by western blotting. The numbers under the blots represent the gray scale quantification (Tom20/Tubulin, Tim23/Tubulin). Source data are provided as a Source Data file.

together, our findings indicate that MCL-1 plays a role in mitophagy.

**MCL-1 is an LC3-interacting mitophagy receptor**. We hypothesized that the UMI-77-induced release of MCL-1 from Bax/Bak allows MCL-1 to interact with LC3, thereby promoting mitophagy. We found that MCL-1 contains three canonical "LC3-interacting region" (LIR) motifs [W/Y/F]XX[I/L/V][18,19] at its C-terminus and that the first two of them (henceforth, LIR[261–264] and LIR[318–321]) are in the cytosolic region of the protein (Fig. 3a, b). We also found that LIR[261–264] is strongly conserved, suggesting it is functional (Supplementary Fig. 4). To test whether UMI-77 induces interaction between MCL-1 and LC3, we performed co-immunoprecipitation assays. As shown in Fig. 3c, the interaction between LC3A and MCL-1 was enhanced, whereas the interaction between Bax and MCL-1 was decreased following UMI-77 treatment. Consistent with this, we also found that overexpression of MCL-1-M (L213A/D218A) which was not able to interact with Bax[26], enhances mitophagy levels (Supplementary Fig. 5a and b). MCL-1 also interacted with other Atg8 family proteins in the presence of UMI-77 (Fig. 3d). To further demonstrate that MCL-1 binds to LC3A in a specific manner, we generated a series of mutations of the MCL-1 LIR[261–264] and LIR[318–321] motifs (W261A, I264A, W261A/I264A, ΔLIR[261–264], F318A, V321A, and F318A/V321A). The mutations of the LIR[261–264] motif of MCL-1, but not that of LIR[318–321] motif, attenuated MCL-1 interaction with LC3A (Fig. 3e). Importantly, a pull-down assay, using proteins purified from a bacterial expression system, revealed that MCL-1 binds to LC3A directly (Supplementary Fig. 6).

Next, we investigated whether UMI-77 enhances the interaction of endogenous MCL-1 and LC3A. Consistent with the notion that the mechanism of mitophagy induction by UMI-77 is via stimulation of the MCL-1-LC3A interaction, the enhancement of this interaction between endogenous MCL-1 and LC3A was observed in situ on mitochondria by using Duolink® PLA technology, following UMI-77 treatment (Fig. 3f, Supplementary Fig. 7). This interaction was also observed at basal levels, in the absence of UMI-77 treatment, reflecting the basal endogenous levels of mitophagy and further, suggesting that MCL-1 is a mitophagy receptor (Fig. 3f).

We then asked whether the interaction of MCL-1 and LC3A has a role in UMI-77-mediated mitophagy activation. As shown in Fig. 3g, h, and Supplementary Fig. 3b, c, MCL-1 knockdown decreased UMI-77-mediated mitophagy in HEK293T-mt-Keima and SH-SY5Y cells, and this defect was rescued by MCL-1 re-

expression of wild-type and LIR[318–321] mutant (F318A/V321A), but not the LIR[261–264] motif mutants.

Taken together, these results demonstrate that MCL-1 is a mitophagy receptor, which directly interacts with LC3A through its LIR[261–264] motif and that this interaction is enhanced by UMI-77, leading to enhanced levels of mitophagy. Moreover, the interaction between MCL-1 and LC3A is critical for UMI-77-mediated mitophagy activation.

**UMI-77 induces mitophagy via the ATG5 pathway, independent of the canonical mitophagy receptor proteins, Bax or Parkin**. We investigated whether other mitophagy receptor proteins play a role in the UMI-77-induced mitophagy or induction of the MCL-1-LC3A interaction. UMI-77 significantly enhanced the interaction between MCL-1 and LC3A, both in wild-type HeLa and HeLa cells with a quadruple knockout of mitophagy receptors NDP52, p62, NBR1, and TAX1BP1 (Fig. 4a, b). This indicated that the induction of mitophagy by UMI-77 was independent of these mitophagy receptor proteins. Consistent with this, we found that the levels of the mitochondrial marker proteins Cox II and Tim23 were decreased by UMI-77 treatment in a time-dependent manner in both wild-type and the quadruple knockout HeLa cells (Fig. 4c). We also found that the previously reported mitophagy receptors (FUNDC1, BNIP3, and NIX) did not participate in UMI-77-induced mitophagy (Supplementary Fig. 8a–d).

Given that UMI-77 blocks the interaction of MCL-1 to Bax[22], we then assayed the role of Bax in the UMI-77-induced mitophagy. Surprisingly, knockdown of Bax enhanced mitophagy levels induced by the UMI-77 treatment (Supplementary Fig. 9a). In contrast, the induction of mitophagy by UMI-77 required ATG5, as knockdown of ATG5 either in HEK293T-mt-Keima cells or in MEF cells prevented Tom20 and Tim23 degradation following UMI-77 treatment (Fig. 4d, e). However, knockdown of Beclin1 did not prevent UMI-77-induced mitophagy, further indicating that UMI-77-induced mitophagy is independent of macroautophagy (Supplementary Fig. 9b, c).

Previous studies have shown that Bcl-2 family proteins inhibit mitophagy by binding to Parkin, and inhibitors of Bcl-2 family proteins reverse this process[27]. However, we observed the induction of mitophagy by UMI-77 treatment in HeLa cells, which has undetectable Parkin protein levels (Fig. 1f, g). These results ruled out the possibility that UMI-77 mediated mitophagy activation by blocking the interaction between MCL-1 and Parkin. Overall, these experiments show that UMI-77-induced mitophagy is mediated via the ATG5 autophagy pathway, independent of the mitophagy receptor proteins NBR1,

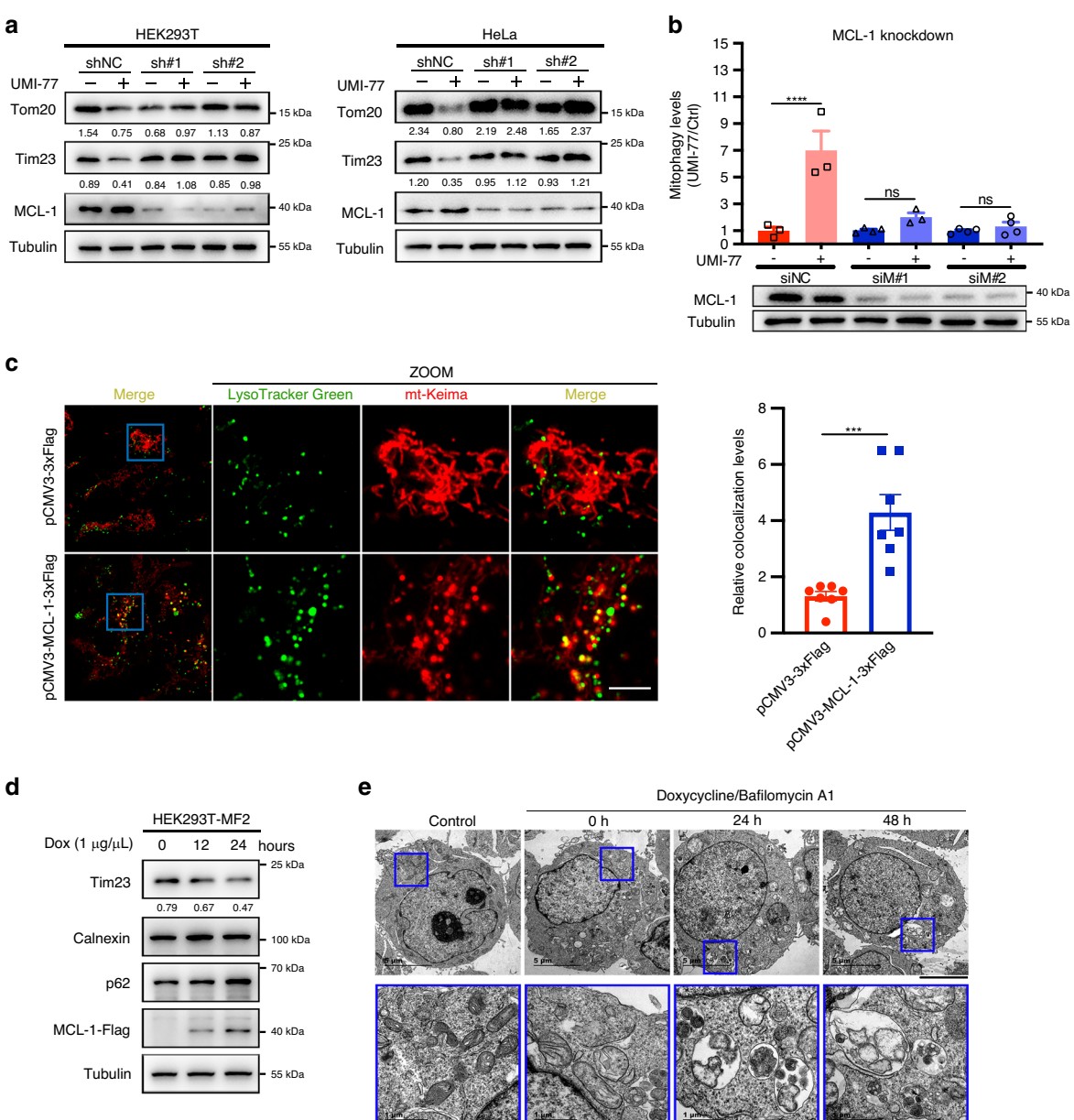

**Fig. 2 MCL-1 promotes mitophagy and is required for UMI-77-induced mitophagy. a** HEK293T cells were transfected with MCL-1 shRNA for 60 h and treated with 5 μM UMI-77 for 12 h. Cell lysates were immunoblotted for mitochondrial marker proteins (Tom20, Tim23). The numbers under the blots represent the gray scale quantification (Tom20/Tubulin, Tim23/Tubulin). shNC: scrambled shRNA. **b** HEK293T-mt-Keima cells were transfected with indicated siRNA for 60 h, treated with UMI-77 (5 μM) for 12 h, The siRNA knockdown efficiency was shown using western blot and the mitophagy levels were quantified. One-way ANOVA (data represent mean ± S.E.M.; $n = 3$. ****$p < 0.0001$, ns, not significant.). siNC: scrambled siRNA. **c** HEK293T-mt-Keima cells were transfected with pCMV3-MCL-1-3xFlag and pCMV3-3xFlag plasmid, for 24 h, stained with LysoTracker Green for 30 min and imaged using fluorescence microscopy. Scale bar, 5 μm. Quantification of the number of LysoTracker-positive dots colocalized with mt-Keima in cells were analyzed by two-tailed $t$ test. (data represent mean ± S.E.M.; $n = 7$ ***$p < 0.001$ ($P = 0.0007$)). **d** MCL-1-expressing HEK293T-MF2 cells were treated with 1 μg/mL doxycycline (Dox) for the indicated times, cell lysates were immunoblotted with indicated antibodies. The numbers under the blot represent the gray scale quantification (Tim23/Tubulin). **e** HEK293T-MF2 cells were treated with 1 μg/mL doxycycline for the indicated times, treated with Bafilomycin A1 for 6 h, and analyzed by transmission electron microscopy (TEM). Insets (blue boxes) show mitochondria and the autophagosomes engulfing mitochondria. Scale bars, 5 μm; insets: Scale bar, 1 μm. Source data are provided as a Source Data file.

TAX1BP1, p62, and NDP52, FUNDC1, BNIP3, NIX, as well as MCL-1 interactor proteins Bax, Beclin1, and Parkin.

**MCL-1 is required for mitophagy induced by oxygen-glucose deprivation.** We then aimed to understand the physiological function of MCL-1 as a mitophagy receptor. Previous studies showed that oxygen-glucose deprivation (OGD) or OGD/reperfusion damage mitochondria and induce mitochondrial clearance through mitophagy[28]. Our mt-Keima assay shows that MCL-1 is required for OGD-induced mitophagy, as knockdown of MCL-1 blocked the excitation shift (Fig. 5a and Supplementary Fig. 3d). The specific degradation of the mitochondrial marker proteins (Cox II and Tim23) induced by OGD was also blocked by knockdown of MCL-1 (Fig. 5b).

MCL-1 has been shown to regulate mitochondrial fragmentation, which is required for mitophagy[29,30]. Therefore, we attempted to

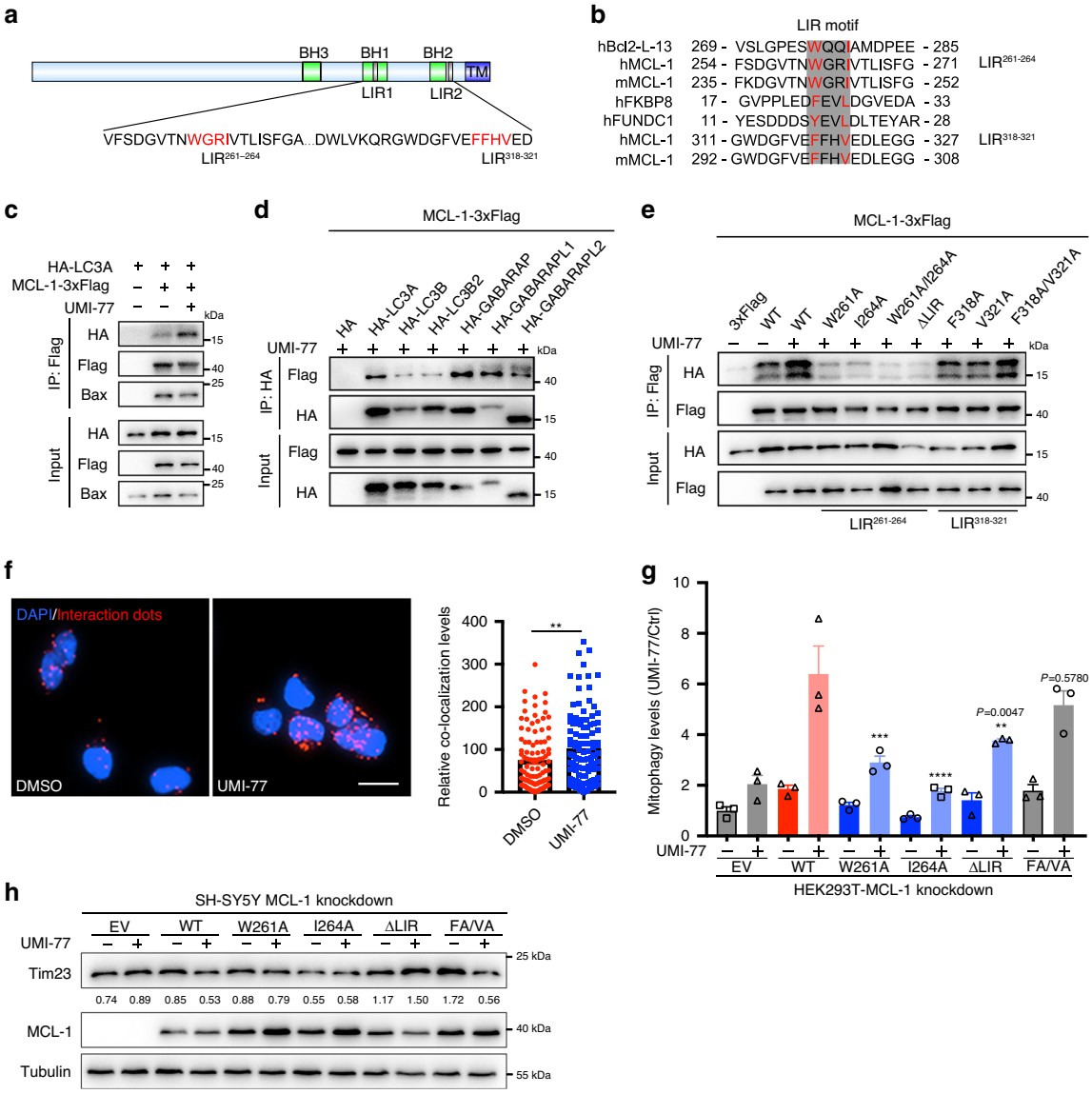

**Fig. 3 The interaction between MCL-1 and LC3A is required for UMI-77-induced mitophagy. a** The schematic diagram of MCL-1 protein indicating two potential LIR (LC3-interacting region) motifs. The image was created by us in this study. **b** The alignment of LIR sequences of MCL-1, Bcl-2-L-13, FKBP8, and FUNDC1. Red sequences indicate highly conserved residues in LIR motif. h: Human; m: Mouse. The image was created by us in this study. **c** HEK293T cells were co-transfected with MCL-1-3xFlag and HA-LC3A for 24 h, treated with UMI-77 (10 μM) for 4 h, and the interaction of MCL-1 with HA-LC3A and endogenous Bax was analyzed by immunoprecipitation. **d** HEK293T cells were co-transfected with MCL-1-3xFlag and HA-tagged Atg8 family proteins for 24 h, treated with UMI-77 (10 μM) for 4 h, and the interaction between MCL-1 and the Atg8 family proteins was analyzed by immunoprecipitation. **e** HEK293T cells were co-transfected with MCL-1-3xFlag WT or the indicated mutants and LC3A-HA for 24 h, treated with UMI-77 (10 μM) for 4 h, and the interactions were analyzed by immunoprecipitation. **f** PLA assay for endogenous MCL-1 and LC3A was performed in HEK293T cells treated with UMI-77 (10 μM) for 4 h. Scale bar, 20 μm. Graph on the right–quantification of the PLA dots (data represents mean ± S.E.M.; DMSO ($n = 107$ cells), UMI-77 ($n = 115$ cells), **$p < 0.01$ ($P = 0.0065$), two-tailed $t$ test). **g** HEK293T-MCL-1-konckdown cell were co-transfected MCL-1 WT or indicated mutants with mt-Keima plasmid for 48 h, treated with UMI-77 (5 μM) for 12 h. The mitophagy levels were quantified by one-way ANOVA (data represent mean ± S.E.M.; $n = 3$, ****$p < 0.0001$, ***$p < 0.001$, **$p < 0.01$, ns, not significant.). **h** SH-SY5Y MCL-1-knockdown cell line was transfected with expression plasmids encoding MCL-1 WT or indicated mutants and treated with UMI-77 (5 μM) for 12 h. The numbers under the blots represent the gray scale quantification (Tim23/Tubulin). Source data are provided as a Source Data file.

understand the role of MCL-1 in OGD-induced mitophagy. Mitochondrial fragmentation was observed in HEK293T cells following OGD, as judged by immunofluorescence microscopy. Notably, knockdown of MCL-1 rescued these morphological changes, indicating that MCL-1 has a critical role in OGD-induced mitochondrial fragmentation (Fig. 5c).

Next, we examined the role of MCL-1 LIR[261–264] motif in the OGD-induced mitochondrial fragmentation and mitophagy. Although overexpression of the LIR[261–264] motif mutants

W261A, I264A, and ΔLIR[261–264] had no effect on the OGD-induced mitochondrial fragmentation in cells with a stable MCL-1 knockdown (Fig. 5d), the OGD-induced mitophagy was significantly blocked by these LIR mutations (Fig. 5e). We also found OGD enhanced the interaction between MCL-1 and LC3A and decreased the interaction with Bax (Fig. 5f). This indicated that MCL-1 acts as a receptor for mitophagy activation during OGD and that the LIR[261–264] motif is only involved in mitophagy, but not in the mitochondrial fragmentation role of MCL-1.

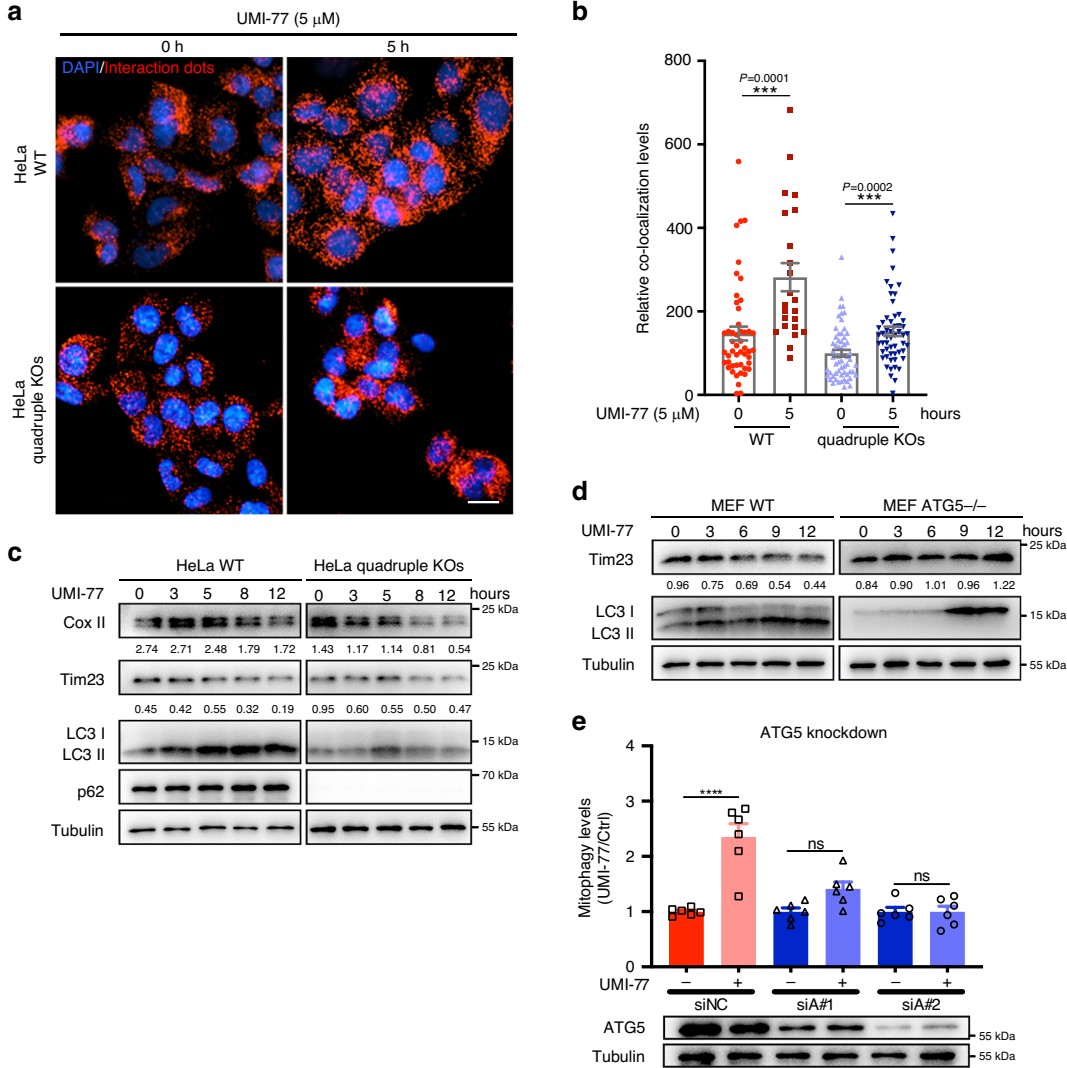

**Fig. 4 UMI-77 induces mitophagy in an ATG5-dependent manner, independent of adapter proteins NBR1, TAX1BP1, p62, and NDP52. a** PLA assay for MCL-1 and LC3A were performed in HeLa WT and quadruple KOs (NBR1, TAX1BP1, p62, and NDP52 knockout) cells treated with UMI-77 (5 μM) for the indicated times. Cells were analyzed by fluorescence microscopy. Scale bar, 20 μm. **b** Quantification of the mean area dots from **a** by two-tailed $t$ test (data represents mean ± S.E.M. The sample size was, in turn, $n = 49$, $n = 23$, $n = 60$, $n = 55$ cells, ***$p < 0.001$). **c** HeLa WT and quadruple KOs (NBR1, TAX1BP1, p62, and NDP52 knockout) cells were treated with 5 μM UMI-77 for the indicated times and cell lysates were immunoblotted with indicated antibodies. The numbers under the blots represent the gray scale quantification (Cox II/Tubulin, Tim23/Tubulin). **d** MEF WT and ATG5 knockout cells were treated with 5 μM UMI-77 for the indicated times, and mitochondrial marker protein Tim23 and LC3 were detected by western blotting. The numbers under the blots represent the gray scale quantification (Tim23/Tubulin). **e** The mitophagy levels of control cells and ATG5 knockdown cells treated with UMI-77 were analyzed using one-way ANOVA (data represent mean ± S.E.M.; $n = 6$. ****$p < 0.0001$. ns, not significant). The siRNA knockdown efficiency was shown using western blot. siNC: scrambled siRNA. Source data are provided as a Source Data file.

**UMI-77 ameliorates cognitive decline and amyloid pathologies in the APP/PS1 mouse model of AD**. Using mt-Keima transgenic mice, we found that intraperitoneal injection of a 10 mg/kg dose of UMI-77 potently induces mitophagy in vivo, in mouse brain tissues, within 6 hours (Fig. 6a). Next, we examined the effect of UMI-77-induced mitophagy on the disease pathologies and mouse behavioral phenotypes of the APP/PS1 mouse model of AD. Each mouse was injected with either vehicle or UMI-77 (10 mg/kg) every other day, from the age of 4 months, for a total period of 4 months. Using the Morris water maze test, we found that the UMI-77 treatment improved the learning and memory of the APP/PS1 mice (Fig. 6b, c). UMI-77 could effectively reduce the levels of the insoluble $A\beta_{1-42}$ in mouse brains (Fig. 6d). Similarly, immunofluorescence results also showed that the size of extracellular Aβ plaque in the hippocampus was significantly reduced, and the activation of astrocytes (as judged by glial fibrillary acidic protein (GFAP) staining) was also inhibited by the UMI-77 treatment (Fig. 6e).

UMI-77 reduced the neuroinflammation levels in the APP/PS1 mice. Inflammatory cytokine levels (TNFα and IL-6) were significantly reduced by the UMI-77 treatment, whereas anti-inflammatory cytokine levels (IL-10) were unaffected (Fig. 6f). Finally, As shown in Fig. 6g, UMI-77 significantly restored the mitochondrial morphology in the neurons, consistent with the notion that induction of mitophagy by UMI-77 would result in the clearance of the damaged mitochondria seen in the APP/PS1 mice.

As our data show that MCL-1 is a mitophagy receptor, next, we attempted to evaluate the effect of MCL-1-induced mitophagy on the behavioral phenotypes of the APP/PS1 mice. Following

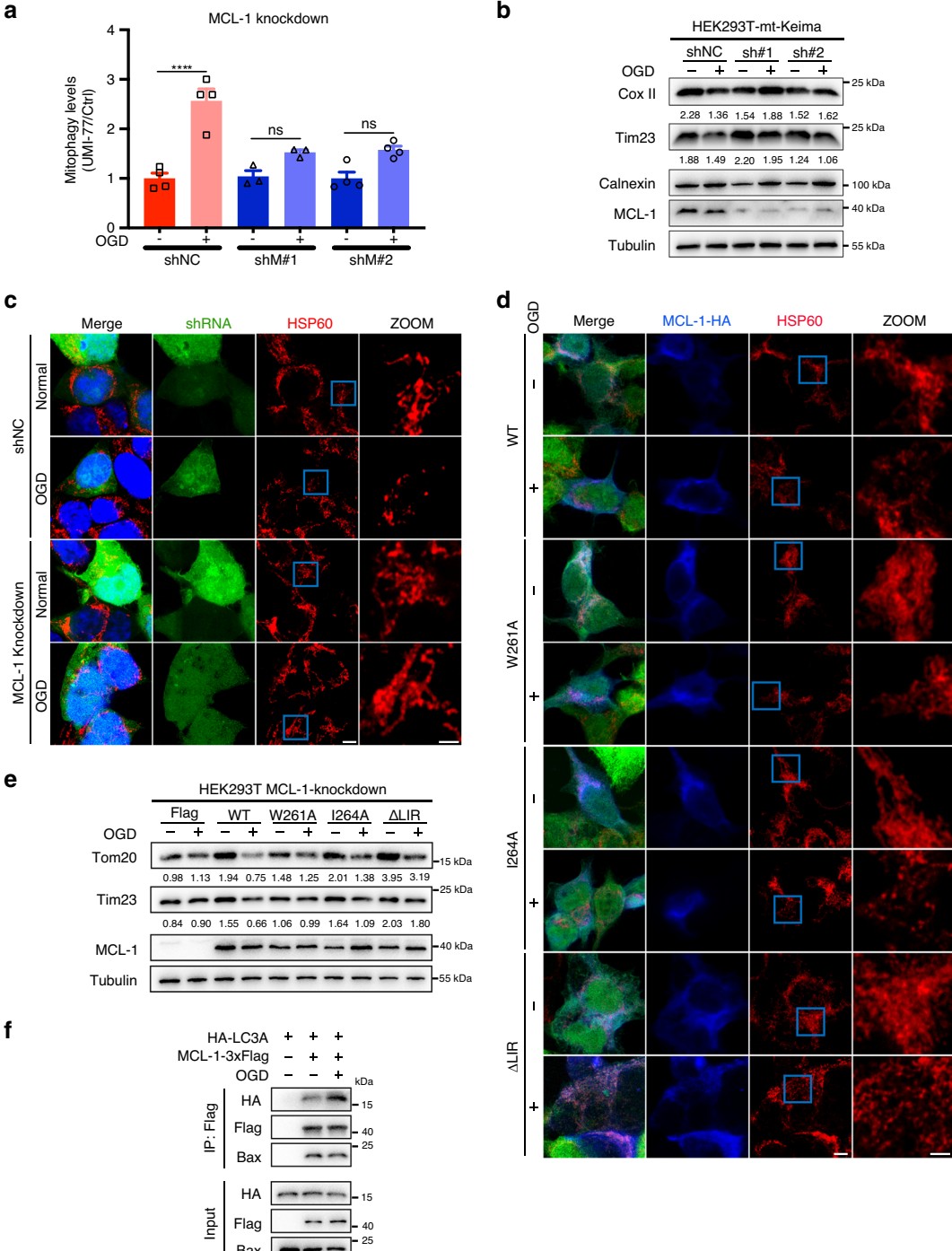

**Fig. 5 MCL-1 is required for mitophagy induced by oxygen-glucose deprivation. a** HEK293T-mt-Keima cells were infected with lentiviral particles encoding MCL-1 shRNA for 24 h and treated with oxygen-glucose deprivation (OGD) for 5 h. Mitophagy levels were analyzed by one-way ANOVA (data represent mean ± S.E.M.; $n = 4$. ****$p < 0.0001$, ns, not significant). shNC: scrambled shRNA. **b** HEK293T-mt-Keima cells were infected with lentiviral particles encoding MCL-1 shRNA for 24 h and treated with OGD for 5 h. Cell lysates were immunoblotted with indicated antibodies. The numbers under the blots represent the gray scale quantification (Cox II/Tubulin, Tim23/Tubulin). shNC: scrambled shRNA. **c** HEK293T cells were infected with lentiviral particles encoding MCL-1-shRNA or control-shRNA for 24 h, treated with OGD for 5 h. Cells were fixed and stained with indicated antibodies. Green channel indicated cells were infected successfully. Scale bar, 5 μm; insets: scale bar, 2 μm. shNC: scrambled shRNA. **d** HEK293T-MCL-1-knockdown cells were transfected with MCL-1 wild-type or indicated mutants for 24 h, treated with OGD for 5 h. Cells were fixed and stained with indicated antibodies, following by fluorescence microscopy analysis. Scale bar, 5 μm; insets: Scale bar, 2 μm. **e** HEK293T-MCL-1-knockdown cells were transfected with MCL-1 WT or indicated mutants for 24 h, treated with OGD for 5 h. Cell lysates were immunoblotted with indicated antibodies. The numbers under the blots represent the gray scale quantification (Tom20/Tubulin, Tim23/Tubulin). **f** HEK293T cells were co-transfected with MCL-1-3xFlag and HA-LC3A for 24 h, treated with OGD for 5 h, and the interaction of Flag-tagged MCL-1 with HA-LC3A and endogenous Bax was analyzed by immunoprecipitation. Source data are provided as a Source Data file.

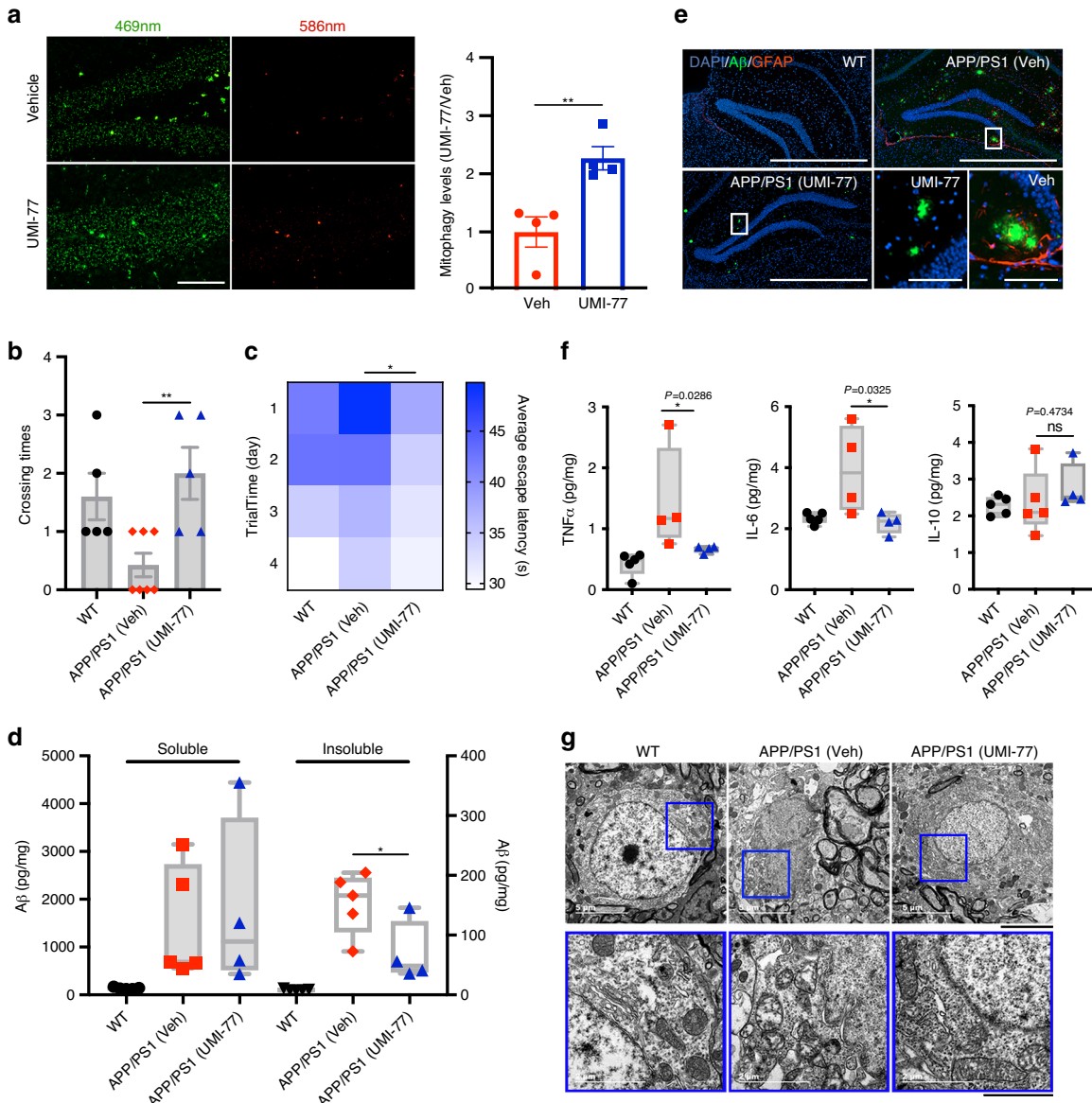

**Fig. 6 UMI-77 ameliorates cognitive decline and amyloid pathologies in the APP/PS1 mouse model of Alzheimer's disease. a** mt-Keima signal induction by UMI-77 in the mt-Keima mouse brain hippocampus samples. Mice were injected with 10 mg/kg UMI-77 for 6 h and samples were analyzed by fluorescence microscopy (**$p < 0.01$ (P = 0.0083), two-tailed $t$ test). Scale bar, 100 μm. **b** The number of times a mouse crossed the platform within 60 s after removing the platform by training with four days ((WT, $n = 5$), (APP/PS1 + Veh, $n = 7$), (APP/PS1 + UMI-77, $n = 5$), **$p < 0.01$ (P = 0.0053), two-tailed $t$ test). **c** Latency to escape to a hidden platform in the Morris water maze during a 4-day training period ((WT, $n = 5$), (APP/PS1 + Veh, $n = 7$), (APP/PS1 + UMI-77, $n = 5$), *$p < 0.05$ (P = 0.0334), two-tailed $t$ test). **d** Mice were treated as in **b** and brain tissues were analyzed for soluble and insoluble $A\beta_{1-42}$ levels, using ELISA (mean ± S.E.M.; *$p < 0.05$ (P = 0.0463), two-tailed $t$ test). Box plots indicate median (middle line), 25th, 75th percentile (box) and minima and maxima (whiskers). **e** Mice were treated as in **b** and IHC of whole brains was performed to stain for amyloid-beta (Aβ) plaques (6E10 antibody, green), astrocytes (GFAP antibody, red) and nuclei (DAPI, blue). Scale bar, 1000 μm; insets: Scale bar, 100 μm. **f** Mice were treated as in **b** and the levels of the indicated cytokine levels were measured by ELISA using whole brain lysates (mean ± S.E.M.; *$p < 0.05$, ns, not significant, two-tailed $t$ test). Box plots indicate median (middle line), 25th, 75th percentile (box) and minima and maxima (whiskers). **g** Electron microscopy images of mice brain hippocampal tissues. Insets (blue boxes) show mitochondria. Scale bars, 5 μm; insets: Scale bars, 2 μm. Source data are provided as a Source Data file.

AAV-mediated delivery of an MCL-1-expressing vector into the hippocampus of these mice, we found that MCL-1 overexpression ameliorates the cognitive decline seen in the APP/PS1 mice and reduces extracellular Aβ plaque in the hippocampus (Supplementary Fig. 10a–c). Surprisingly, overexpression of MCL-1 also improved the learning and memory of wild-type mice, indicating that MCL-1 has an important role in neurons (Supplementary Fig. 10a).

In conclusion, UMI-77 potently induced mitophagy in vivo, significantly restored cognitive deficits of the APP/PS1 mouse model of AD, reduced the inflammatory response, and the pathological effects caused by the Aβ plaques, and promoted clearance of the damaged mitochondria. Moreover, confirming our UMI-77 findings, overexpression of MCL-1 in the hippocampus of the APP/PS1 mice phenocopied these results. Overall, these experiments suggest that UMI-77 is a potent drug lead for the treatment of AD.

## Discussion

Our study shows that MCL-1, an important anti-apoptotic protein, is a LC3-interacting mitophagy receptor protein that induces mitochondrial fragmentation and mitophagy in response to mitochondrial damage caused by OGD. Our results suggest that MCL-1 mediates mitochondrial fragmentation and mitophagy through distinct molecular mechanisms. Although the mitophagy role of MCL-1 requires its interaction with LC3, the mitochondrial fragmentation role of MCL-1 is independent of this interaction.

We postulate that MCL-1 recruits LC3A via its LIR motif to the surface of mitochondria, leading to the formation of nascent mitophagosomes and the elongation of the mitophagosome membrane. The interaction between MCL-1 and GABARAP proteins subsequently mediates the closure of the mitophagosome membrane, thereby engulfing the mitochondria. Consistent with this, our data showed that MCL-1 can interact with both LC3A and GABARAP proteins (Fig. 3c, d). However, we are not able to rule out the possibility that MCL-1 can cooperate with NIX and/or FUNDC1 for mediating mitophagy. As both NIX and MCL-1 specifically interact with LC3A, there may be a synergistic mechanism between NIX and MCL-1 to promote mitophagy induction[31].

Previous studies suggest that loss of MCL-1 in adult myocytes results in mitochondrial dysfunction, defective PINK1-PARK2 signaling and impaired initiation of autophagy[32]. MCL-1 is also suggested to function in embryonic development and synapse formation[33,34], lifespan regulation[35,36], and diverse cellular processes (including mitochondrial dynamics)[32,36,37]. Our findings thus provide insights into how MCL-1 can act as a mitophagy receptor to couple with the molecular mechanisms of mitophagy and the mechanisms by which MCL-1 regulates normal physiology, aging, and disease.

We also noticed that MCL-1 is reported to act as a suppressor of AMBRA1 to suppress mitophagy, under the conditions of FCCP or CCCP treatment[38]. It is reported that AMBRA1-mediated mitophagy is regulated by HUWE1, an E3 ubiquitin ligase, which could control mitochondrial protein ubiquitylation in cooperation with AMBRA1 during the mitophagy process in a PARKIN-free cellular system, in the condition of MMP deprivation[39]. And MCL-1 overexpression is sufficient to inhibit recruitment to mitochondria of HUWE1, upon AMBRA1-mediated mitophagy induction[38]. However, PINK1/PARKIN, but not HUWE1/AMBRA1, is the main pathway regulating mitophagy in cells with PARKIN, following MMP deprivation, during which the inhibition of MCL-1 on HUWE1 translocation to mitochondria may not inhibit mitophagy. What's more, these previous findings suggest that MCL-1 may inhibit ubiquitin-dependent mitophagy through inhibition of E3 ligase translocation to depolarized mitochondria, while our findings suggest that MCL-1-mediated mitophagy may be in a ubiquitin-independent way.

We found that several apoptosis inducers can induce mitophagy to an extent much weaker than UMI-77. Among them, Gambogic acid inhibits most of the Bcl-2 family proteins, including MCL-1, whereas AT101 inhibits Bcl-2, Bcl-xL, and MCL-1, suggesting that the mechanism of mitophagy induction by them could be similar to that of UMI-77. ABT-737 and ABT-263 are BH3-mimetics, similar to UMI-77, but they have not been reported to inhibit MCL-1, suggesting that other Bcl-2 family proteins may participate in the regulation of mitophagy. In addition, ABT-737 has been reported to promote mitophagy via inducing Parkin translocation to mitochondria by inhibiting the interaction between Bcl-2 and Parkin. RITA promotes p53 phosphorylation to induce apoptosis, whereas GDC-0152 inhibits the interaction between IAP and other proteins. Notably, both

IAP and p53 have been suggested to be involved in the regulation of mitophagy[27,40–44].

Mitophagy induction is a promising strategy for AD therapy[1,11]. CCCP and oligomycin A are two wide-used mitophagy-inducing agents that drive mitophagy by causing mitochondrial damage[45,46]. However, their severe cytotoxicity limits their clinical application. Nicotinamide riboside (NR) is a safe and effective activator of neuronal mitophagy[47]. NR is a precursor of $NAD^+$ and can be metabolized to produce $NAD^+$ in cells which can rescue the function of mitochondria in xeroderma pigmentosum group A[48]. NR reduces Aβ levels in APP/PS1 mice and has been tested in clinical trials[47,49]. Urolithin A (UA) is a natural, dietary, microflora-derived metabolite, which can induce mitophagy and ameliorate cognitive decline in the APP/PS1 mouse model[11,50,51].These studies highlight the necessity to identify more safe, effective, and clear mechanisms for mitophagy inducers that can become effective therapeutic approaches for the treatment of AD. Our discovery of the FDA-approved drug candidate MCL-1 BH3-mimetic UMI-77 is an unexpected mitophagy-inducing agent that does not cause mitochondrial damage. Previous studies showed that UMI-77 can induce apoptotic cell death in certain cancer cell lines that express high levels of MCL-1[22,37]. However, using unbiased systematic approaches, we found that UMI-77 can induce mitophagy fully independent of apoptosis and that it can ameliorate the pathologies seen in the APP/PS1 mice.

Risk factors in AD patients, such as genetic mutation, aging, and environmental factors, promote the production of reactive oxygen species (ROS) and lead to mitochondrial dysfunction, as well as the production of Aβ plaques. The abnormal mitochondrial function accelerates the production of ROS and fueling this process. Therefore, degradation of damaged mitochondria by mitophagy can prevent excessive production of ROS, reducing the production of Aβ production[52]. On the other hand, Aβ can be transported into cells and accumulated in mitochondria, and this interaction inhibits mitochondrial function, elevates ROS levels, and alters mitochondrial dynamics[53–55]. We report that UMI-77-induced mitophagy effectively reduces Aβ plaque levels, which may be achieved through decreased production of ROS and the Aβ deposited in the damaged mitochondria. Our work strongly supports UMI-77 as a putative drug for the treatment of AD.

In summary, we identify MCL-1 as a mitophagy receptor that can be targeted by the FDA-approved drug candidate BH3-mimetic UMI-77 to induce mitophagy and promote reversal of the AD pathology. Our findings suggest MCL-1 as a drug target for AD and further confirm that induction of mitophagy is a viable strategy for treating this neurodegenerative disorder. MCL-1 is an important cancer drug target, and several clinical trials are currently underway[56]. Moreover, UMI-77 is an FDA-approved drug candidate for the treatment of pancreatic cancer[22]. In addition, our discovery of the role of MCL-1 as a mitophagy mediator and the ability of MCL-1 inhibitors to induce mitophagy may help to consider this function of the anti-apoptotic protein when designing and evaluating cancer therapies that target MCL-1.

## Methods

**Reagents and antibody generation**. UMI-77 (#T6034) was purchased from Topscience, Inc. (Shanghai, China). MG-132 (#S2619), bafilomycin A1 (#S1413), leupeptin hemisulfate (#S7380), E64D (#S7393) were from SelleckChem, Ltd. Doxycycline (#A603456) and NH4Cl (A501569) were purchased from Sangon® Biotech, Inc. (Shanghai, China). The following antibodies were used: anti-Tom20 (1:1000, #42406 S, Cell Signaling Technology, Inc.), anti-Tim23 (1:1000, #11123-1-AP, Proteintech, Inc.), anti-Calnexin (1:1000, #2433 S, Cell Signaling Technology, Inc.), anti-MCL-1 (1:200, #sc-698401, Santa Cruz Biotechnology, Inc.), anti-p62 (1:1000, #18420-1-AP, Proteintech, Inc.), anti-Tubulin (1:5000, #M1305-2, HUA-BIO, Ltd.), anti-Actin (1:5000, #M1210-2, HUABIO, Ltd.), anti-GAPDH (1:2000,

#ET1601-4, HUABIO, Ltd.) anti-Flag (1:5000, #M1403-2, HUABIO, Ltd.), anti-HA (1:5000, #0906-1, HUABIO, Ltd.), anti-LC3A (1:1000, #ab62720, Abcam, Ltd.), anti-Cox II (1:1000, ET1610-72, HUABIO, Ltd.), anti-LC3 (1:2000, #L8918, Sigma, Ltd.). The secondary antibodies for western blot were used: goat anti-mouse (1:20000, #31430, Thermo Fisher Scientific, Ltd.), goat anti-rabbit (1:20000, #31460, Thermo Fisher Scientific, Ltd.). The fluorescent secondary antibodies for immunofluorescence were used: goat anti-mouse Alexa Fluor 488 (1:500, A-11001, Thermo Fisher Scientific, Ltd.), goat anti-rabbit Alexa Fluor 546 (1:500, A-11010, Thermo Fisher Scientific, Ltd.), goat anti-rabbit Alexa Fluor 555 (1:500, A-21428, Thermo Fisher Scientific, Ltd.), goat anti-mouse Alexa Fluor 405 (1:500, A-31553, Thermo Fisher Scientific, Ltd.).The antibodies for Immunohistochemistry (IHC) were used: Aβ (1:50, Biolegend, Ltd.), GFAP (1:50, DAKO, Ltd.). The following beads were used for Immunoprecipitation: Anti-Flag (DYKDDDDK) Affinity Gel (#B23102) and Anti-HA magnetic beads (#B26202) were purchased from Bimake, Ltd.

**High-throughput screening**. FDA-approved drug or drug candidates library was purchased from Topscience, Inc. (Shanghai, China). Phosphate-buffered saline (PBS) was used as a positive control to induce mitophagy[57,58], whereas dimethyl sulfoxide (DMSO) was used as a negative control, since compounds from the screen were dissolved in DMSO. Mitophagy inducers were screened using HEK293T-mt-Keima stable cell line. High-throughput imaging was done using Biotek Cytation® 3 system excitation at 469 nm/586 nm and emission at 620 nm. Mitophagy levels were estimated by the fraction of cells that were fluorescent upon excitation at 586 nm using Gen5™ software. The screen Z factor was >0.5. A compound with excitation shift fluorescence intensity value higher than negative control by 1.5-fold was scored as a mitophagy inducer.

**Cell culture**. HEK293T, SH-SY5Y, MEF WT, MEF ATG5$^{-/-}$, HeLa, HeLa quadruple KOs (NBR1, TAX1BP1, p62, and NDP52 knockout), U2OS, HEK293T-MCL-1-knockdown and SH-SY5Y-MCL-1-knockdown cell lines were grown in DMEM medium (Hyclone™, with L-glutamine, with 4.5 g/L glucose, without pyruvate) at 37 °C under 5% CO$_2$. These media were supplemented with 10% fetal bovine serum (FBS; Gibco™), 1% penicillin/streptomycin (Gibco™). For oxygen-glucose deprivation, cells were grown in DMEM medium without glucose (#11966025, Thermo Fisher Scientific, Ltd.) at 37 °C under gas mixture of 1% O$_2$, 5% CO$_2$ and 95% N$_2$. Doxycycline-inducible HEK293T-MF2 stable cell lines were generated by co-transfecting HP138-MCL-1 and HP216 plasmids (a gift from Dr. Hui Yang) into HEK293T cells using Lipofectamine™ 2000 (Invitrogen™) and selecting with 10 μg/mL puromycin (Sangon® Biotech, Inc.). The mt-Keima gene sequence was synthesized and cloned into pCDNA3.1 by Shanghai Tologo Biotechnology, China.

The primers pCDH-mt-Keima-F 5′-CCGGAATTCGAAATGCTGAGCCTG CGCCAGAG-3′ and pCDH-mt-Keima-R 5′-CGCGGATCCTCAACCGAGCAAA GAGTGGC-3′ were used for cloning mt-Keima into the pCDH vector. Lentiviral packaging was done according to previously established methods[59]. Packaging vectors pSPAX2 and pMD2.0 G were co-transfected with pCDH-mt-Keima into HEK293T-mt-Keima cells for 48 h. The cell supernatant was collected and filtered with 0.22-micron filter membrane. Then the virus was concentrated with 4% PEG8000 and 3% NaCl solution overnight. After centrifugation at 5000 rpm, discard the supernatant, add appropriate amount of PBS to resuspend the virus, and store it at – 80 °C. HEK293T-mt-Keima stable cell lines were generated using lentiviral infection for 48 h and selection with 2 μg/mL puromycin. Clones with high mt-Keima expression levels were isolated by flow cytometry.

**Molecular cloning and siRNA-mediated knockdown**. MCL-1-Flag, HA-LC3A, HA-LC3B, HA-LC3B2, HA-GABARAP, HA-GABARAPL1, HA-GABARAPL2 were cloned into pCDNA3.1 via BamHI and HindIII restriction enzymes by PCR amplifying the ORFs from cDNA templates of MCL-1, LC3A, LC3B, LC3B2, GABARAP, GABARAPL1, GABARAPL2, which was provided by Life Sciences Institute, Zhejiang University. Molecular cloning was performed using T4 DNA ligase and transformation into DH5α Escherichia coli cells. Plasmid purifications and extractions were performed using the NucleoBond® Xtra Midi kit (Macherey-Nagel). siRNAs were transfected into HEK293T and HeLa cells using Lipofectamine™ 2000 (Invitrogen™), according to manufacturer's protocol. The siRNA target sequences for BNIP3 knockdown were 5′-GGAAAGAAGTTGAAAGCATC T-3′ and 5′-GGAACACGAGCGTCATGAAGA-3′. The siRNA target sequences for NIX knockdown were 5′-GGATGCACAACATGAATCATT-3′ and 5′-GATCA TGTTTGATGTGGAAAT-3′ The siRNA target sequences for FUNDC1 knockdown were 5′-CCTGAAATCAACAATTTAATT-3′ and 5′-GCACCTGAAATCAA CAATATT-3′ The siRNA target sequences for Beclin1 knockdown were 5′-GGTC TAAGACGTCCAACAACA-3′ and 5′-GGATGACAGTGAACAGTTACA-3′. The siRNA target sequences for ATG5 knockdown were 5′-CATCTGAGCTACCCGG ATATT-3′ and 5′-GAAGGTTATGAGACAAGAAGA-3′. The siRNA target sequences for Bax knockdown were 5′-GGGACGAACTGGACAGTAACA-3′ and 5′-GAACTGATCAGAACCATCATT-3′.

**shRNA-mediated knockdown**. The shRNA sequences were cloned into the pLV3 vector. Packaging of lentiviruses was performed according to previous methods[59].

HEK293T or SH-SY5Y cells were infected with the lentiviral particles for 48 h and selected with 2 μg/mL puromycin. The shRNA target sequences for MCL-1 knockdown were: 5′-GCAGGATTGTGACTCTCATT-3′ and 5′-AGGCTTGCTTGTTACACAC-3′. The non-targeting control (NC) "scrambled" (shNC or siNC) shRNA and siRNA sequence was 5′-TTCTCCGAACGTGAT-CACGTTT-3′.

**Mitochondrial membrane polarization**. HEK293T and HeLa cells were treated with UMI-77 (5 μM) or CCCP (10 μM) for 12 h. Cells were stained by JC1 according to operation manual. Depolarized mitochondria levels were analyzed by Biotek Cytation® 3.

**Immunoprecipitation**. Cells were cultured in 10-cm culture dishes, transfected as described above, washed twice with 5 ml PBS and scraped into 1 ml of pre-chilled RIPA buffer (20 mM Tris-HCl, 150 mM NaCl, 0.5% NP-40, 1 mM NaF, 1 mM Na$_3$VO$_4$, 1 mM EDTA) plus protease inhibitor cocktail. Lysates were incubated for 30 min at 4 °C. After 15,000 × g centrifugation for 10 min at 4 °C, the lysates were subjected to immunoprecipitation with anti-Flag or anti-HA agarose beads, overnight at 4 °C. Beads were collected and washed three times with 1 ml RIPA buffer. The complexes were eluted with 2× sodium dodecyl sulfate polyacrylamide gel electrophoresis (SDS-PAGE) loading buffer for 10 min at 100 °C.

**Immunoblotting**. Samples were heated at 100 °C for 10 min, subjected to 10–12% SDS-PAGE, and transferred onto polyvinylidene difluoride membranes for 1 h at 0.2 A in a wet transfer tank submerged into an ice bath. Membranes were blocked in PBS with Tween (PBST) buffer containing 5% (w/v) skimmed milk for 1 h and probed with the indicated antibodies in PBST containing 5% (w/v) BSA at 4 °C overnight. Detection was performed using HRP-conjugated secondary antibodies and chemiluminescence reagents (#4 AW001-500, Beijing 4A Biotech Co., Ltd.).

**Protein purification and GST-pull down**. The truncated MCL-1 (1-325aa) was cloned into the pET28a vector and transformed into E. coli BL21 (DE3). The primers for pET28a-MCL-1 and GST-LC3A were shown in Supplementary Table 1. Protein expression was induced by 1 mM IPTG addition. Proteins were purified using Ni-NTA columns. GST-LC3A expression was induced using the same method and the protein was purified using glutathione-Sepharose. GST-LC3A was incubated with purified MCL-1 for 2 h at 4 °C. GST beads were washed four times with lysis buffer. Proteins were eluted at 100 °C for 10 min with 20 μl SDS-PAGE loading buffer and analyzed by western blotting.

**Cell viability assay**. HEK293T and SH-SY5Y cells were cultured in 96-well plate for 24 h and treated with UMI-77 for 12 h. Cells were stained with LIVE/DEAD™ cell imaging kit (#R37601, Thermo Fisher Scientific, Ltd.) according to operation manual. The images were obtained from Biotek Cytation® 3 system by using 488 nm (indicate live cell) and 586 nm (indicate dead cell) excitation. The cell viability was estimated by ratio of live cells. HEK293T and SH-SY5Y cells were cultured in 96-well plate for 24 h and treated with UMI-77 for 24 h. Caspase-3 activity was determined by using Caspase-Glo® 3/7 Assay System kit (#G8090, Promega). The data were obtained from Biotek Cytation® 3 system.

**Proximity ligation assays (PLA)**. Cells were cultured on glass slides, treated with or without UMI-77, washed twice with PBS, and fixed with 4% paraformaldehyde in PBS for 20 min at 25 °C. Following blocking with 5% FBS supplemented with 0.1% Triton X-100 to increase the permeabilization for 1 h, primary antibodies (MCL-1 and LC3A) were incubated with the slides overnight at 4 °C. After incubating with the secondary antibodies conjugated with the PLA probes, the signals were amplified through the ligation and amplification steps. The fluorescence analysis was done using Biotek Cytation® 3.

**Fluorescence microscopy**. Cells were cultured in 12-well plates on circular glass coverslips and washed twice with PBS and fixed with 4% paraformaldehyde in PBS for 30 min at 25 °C. Cells were blocked with blocking buffer (5% FBS, 0.1%Triton X-100, PBS) at 25 °C for 1 h, and incubated with primary antibodies overnight at 4 °C. After washing with PBS, secondary antibodies were incubated at room temperature for 1 h. For mitochondria imaging, cells were stained with Mito-Tracker™ Deep Red FM (#M22426, Thermo Fisher Scientific, Ltd.). For live-cell imaging, cells were cultured on a 3.5 cm glass dish and stained with LysoTracker™ Green DND-26 (#L7526, Thermo Fisher Scientific, Ltd.) and imaged with Zeiss LSM 880 AiryScan or Nikon A1 confocal imaging system. Images were processed with ImageJ software.

**Immunohistochemistry (IHC)**. The IHC of mouse hippocampal tissues was done by HUABIO, China, with mouse anti-β-Amyloid, 1-16 antibody (clone 6E10, #803002, Biolegend, Ltd., 1:100), polyclonal rabbit anti-GFAP antibody (#Z033401-2, DAKO, Ltd., 1:100) imaging with Biotek Cytation® 3.

**Electron microscopy**. Cells were imaged at the Center of Cryo-Electronic Microscopy Zhejiang University using Tecnai™ G2 Spirit. Cells were fixed with glutaraldehyde.

**Animal work**. Male transgenic APP/PS1 (C57BL/6) mice at the age of 5–6 weeks were purchased from Model Animal Research Center of Nanjing University (Nanjing, Jiangsu, China) and housed in a pathogen-free environment of Animal Experimental Center of Zhejiang University. All animal studies and experimental procedures were approved by the Animal Care and Use Committee of the animal facility at Zhejiang University. Mice were divided into three groups: normal C57 group (WT, $n = 5$), control APP/PS1 group (APP/PS1 + Veh, $n = 7$), and APP/PS1 treated with UMI-77 group (APP/PS1 + UMI-77, $n = 5$). The mice in the UMI-77 group were injected intraperitoneally with 10 mg/kg dose of UMI-77 dissolved in buffer (2% DMSO, 30% PEG 400) every 2 days from 16 weeks of age. For MCL-1 overexpression mouse, 6-month-old mice were divided into six groups: normal C57 group (WT, $n = 6$), C57 AAV control group (WT (Ctrl vector), $n = 5$), C57 AAV MCL-1 group (WT (MCL-1 overexpression), $n = 6$), APP/PS1 group (APP/PS1, $n = 6$), APP/PS1 AAV control group (APP/PS1 (Ctrl vector), $n = 6$), APP/PS1 AAV MCL-1 group(APP/PS1 (MCL-1 overexpression), $n = 5$). The mouse MCL-1 gene was synthesized, cloned and packaged into adeno-associated virus (AAV) by Vigene biosciences, China. Stereotactic injections were performed in anaesthetized mice, and virus was injected 1.5 µl at a rate of 0.3 µl every 2 minutes (A/P: −2.0 mm, M/L: −1.5 mm, D/V: −1.5 mm), as described[60]. After 30 days, the mice were performed for Morris water maze assay. Transgenic mt-Keima report mouse was generated according to pervious study by inserting a single copy of mt-Keima into Hipp1 locus on chromosome 11[20].

**Morris water maze**. The circular pool containing titanium dioxide water is divided into four quadrants (northwest, northeast, southwest, and southeast). The platform (12 cm) is placed about 1 cm below the Southeast quadrant water. The mice are put into the water with their heads facing the wall of the pool. The mice were put into the pool four times from different quadrants. The time that each mouse spends to find the underwater platform was recorded. If the time is more than 60 s, the mice are guided to the platform and allowed to stay for 20 s. Moreover, the time to find the platform was recorded as 60 s. Each mouse was trained four times per day for a total of four days, and the time to find the platform (escape latency) was recorded. After training for 24 h, the platform was removed, and the space search experiment was started for 60 s. The mice were put into the water from the opposite side of the original platform quadrant, and the number of times the mice passed through the original platform position and the locus of the movement were recorded.

**ELISA measurements for Aβ and cytokines**. Mouse brain tissues were prepared according to the brain tissue homogenate protocol described in the enzyme-linked immunosorbent assay (ELISA) technical guide from Life Technologies. The accumulation of $A\beta_{1-42}$ was quantified by ELISA (# KHB3441, Thermo Fisher Scientific, Ltd.). ELISA kits were also used to measure the levels of TNFα (# BMS607-2, Thermo Fisher Scientific, Ltd.), IL-6 (# BMS603-2, Thermo Fisher Scientific, Ltd.), and IL-10 (# 431417, Biolegend, Ltd.).

**Statistics and reproducibility**. $P$ values were computed using two-tailed $t$ test or one-way analysis of variance. Data were analyzed using GraphPad Prism 8. All the western blot, micrographs assay, mitophagy levels and apoptosis assay were carried out at least three independent times with the same results.

**Reporting summary**. Further information on research design is available in the Nature Research Reporting Summary linked to this article.

## Data availability
The data that support the findings of this study are available from the corresponding author upon reasonable request. Source data are provided with the paper.

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

## Acknowledgements

Financial support from the National Natural Science Foundation of China (No. 91854108, 81773182 and 31601121) and the National Key R&D Program of China (2017YFA0104200) is gratefully acknowledged. We thank professor Hui Yang from Institute of Neuroscience, Chinese Academy of Sciences for transgenic mt-Keima reporter mouse, professor Michael Lazarou from Monash University and professor Hanming Shen from National University of Singapore for HeLa WT and NDP52, p62, NBR1, and TAX1BP1 quadruple knockout cells. We thank the Imaging Center of Zhejiang University School of Medicine for assistance with confocal microscopy. We thank Chenyu Yang in the Center of Cryo-Electron Microscopy (CCEM), Zhejiang University for her technical assistance on TEM.

## Author contributions

H.X. conceived and coordinated the project. H.X. and A.N. designed the experiments. J.W., A.N. and H.X. interpreted the data and wrote the manuscript. X.C., Y.C., X.X., R.W. and F.H. performed most of the experiments. Q.Z., Q.S., C.Y. assisted with the experiments and helped to analyze the data.

## Competing interests

X.C. and H.X. have filed a patent covering the potential application of UMI-77 in AD. The other authors declare no competing interests.
