## [Peer Review File · Nature Communications]

Reviewers' comments:

Reviewer #1 (Remarks to the Author):

In the research article 'Targeting MCL-1 promotes mitophagy and ameliorates Alzheimer's disease pathologies', the authors show two novel findings, including 1) MCL-1 is a mitophagy receptor and directly binds to LC3A and interacts with ATG5, and 2) UMI-77 can bind on MCL-1 and UMI-77 ameliorates cognitive decline and amyloid pathologies in the APP/PS1 mouse model.

The authors had very detailed and series biochemistry, cell biology-based data and mouse experiments and with impressive amount of data. There are a couple of major concerns from this reviewer that the authors should consider:

1. Whether mitophagy plays a role in UMI-77-induced inhibition of AD memory loss and pathology in mice? It is convincing from the data that UMI-77 can induce mitophagy, and UMI-77 can forestall AD pathology. However, the current data do not provide direct evidence that UMI-77 forestalls AD through mitophagy. Although mitophagy induction can inhibit AD, it does not exclude possibilities that UMI-77-based non-autophagy pathway(s) may also inhibit AD. Some suggested experiments include to check whether overexpression of MCL-1 inhibits memory loss and pathology in the APP/PS1 mice. It may be too long to generate an MCL-1 overexpressed mouse strain and cross with the APP/PS1 mice; but it is capable that the authors to use AAV to overexpress Mcl-1 in hippocampus or the entorhinal cortex of the APP/PS1 mice, and to follow up behavioral and pathological studies. Or

2. Can UMI-77 also induce macro-autophagy at the dose it induces mitophagy? (E.g., the authors mentioned 'sub-lethal doses, UMI-77 potently induces mitophagy'). Since 'UMI-77-induced mitophagy is mediated via the ATG5 autophagy pathway', if it interacts with the general autophagy protein ATG5, then UMI-77 will very likely to induce macroautophagy. If UMI-77 induces both mitophagy and macroautophagy, it is fine; the authors just need to provide the data.

3. Mitophagy inhibition vs. mitophagy induction. This reviewer understands that MCL-1 induces mitophagy via LC3A in the condition of oxygen-glucose deprivation, while MCL-1 also inhibits mitophagy via AMBRA1 inhibition in the condition of MMP deprivation. But the molecular mechanisms on how MCL-1 balances mitophagy induction and mitophagy inhibition are still not clear.

Minor comment

1. 'However, no safe and effective mitophagy activators are available', 'Though mitophagy is suggested as a potential therapeutic approach for AD, there is a lack of safe and effective pharmacological compound that can cross blood-brain barrier and selectively induce mitophagy'. These statements are not correct. The NAD⁺ precursor nicotinamide riboside (NR) induces neuronal mitophagy and is safe based on the data from over 5 clinical trials (PMID: 31577933). And another anti-AD mitophagy inducer urolithin A is safe based on the evidence in a recent clinical trial (<https://www.nature.com/articles/s42255-019-0073-4>). Such studies should be mentioned in the discussion session.

Reviewer #2 (Remarks to the Author):

The authors screened a library of FDA-approved drugs or drug candidates, and identified UMI-77, a previously reported BH3-mimetic to induce apoptosis, as a novel mitophagy activator at sublethal doses. They showed that MCL-1 is LIR motif to directly bind to LC3 and to mediate mitophagy in response to oxygen-glucose deprivation in an ATG-5 dependent manner. The authors also claimed that UMI-77 can induce mitophagy in vivo and ameliorate the AD phenotypes of APP/PS1 mice. The

authors provide interesting set of data to show UMI-77 activates Mcl-1 dependent mitophagy. However, they do not show how MCL-1 is activated by UMI-77 to induce mitophagy. Does its inhibition on the interaction between MCL-1 and Bax/Bak lead to the exposure LIR of MCL1 for mitophagy? It would be important to dissect the interplay between apoptosis and mitophagy as the LIR motif is within the BH1 domain of Mcl-1. Also, strong evidence and clearer images are needed to show the mitophagy induction ability in vivo (in brain) by UMI-77.

Specific comments include:

1. In the paper, the cells were treated with UMI-77 for only 12 hours, and then the cell viability assay was performed. The treatment duration should be extended to at least 24 hours to determine the if the drug is lethal at 5 μ M. Also, Live/dead kit is used in the paper to stain dead cells with damaged cell membranes, which is a later apoptosis event. It is known that UMI-77, as low as 3 μ M, can induce caspase 3 cleavage in pancreatic cancer cells. (PMID: 24019208).
2. Other mitophagy receptors, such as BNIP3, FUNDC1, NIX, should be checked if they also are involved in UMI-77 induced mitophagy, as these receptors have been suggested to participate in hypoxia induced mitophagy.
3. In addition to mitochondria, MCL1 also localizes at the endoplasmic reticulum, the Golgi apparatus and the peroxisomes. Why does UMI-77 treatment only induce mitophagy?
4. Images in Figure 5e are not convincing.
5. Figure 4d: the LC3 bands in the right panel are weird. UIM-77 induces autophagy in ATG5 KO cells?
6. Figure 6f: the mice number of APP/PS1(veh) set is 5 when the level of IL-10 was measured, while the number is 4 when TNF-alpha and IL-6 were measured. Please explain

Reviewer #3 (Remarks to the Author):

In this study, the authors identified UMI-77 as a novel mitophagy inducer by screening FDA-approved drugs, and UMI-77-induced mitophagy is mediated by MCL-1. Besides, the mitophagy induction by UMI-77 improved the pathogenic phenotypes in model mice for Alzheimer's disease. This study is novel and interesting. However, the authors need to perform experiments to confirm their conclusion.

1. The authors suggest that the release of MCL-1 from Bax/Bak allows MCL-1 to interact with LC3, thereby promoting mitophagy. To confirm this, the authors should show the mitophagic activity of MCL-1 mutant, which cannot interact with Bax/Bak. Also, the authors should show that UMI-77 or OGD inhibits the interaction between MCL-1 and Bax/Bak. In addition, the authors should show OGD enhances the interaction between MCL-1 and LC3A.
2. MCL-1 is reported to inhibit autophagy mediated through Beclin-1 interaction. Is MCL-1-induced mitophagy Beclin-1-dependent?
3. Fig2c-e show overexpression of MCL-1 induced mitophagy in the absence of UMI-77. However, in Fig 3c, overexpression of MCL-1 did not induce LC3A-MCL-1 interaction in the absence of UMI-77. In Fig 3g, the mitophagy levels were not different between WT and mutants in the absence of UMI-77. These indicate overexpression of MCL-1 does not induce mitophagy in the absence of UMI-77. How do the authors reconcile the apparent discrepancy?
4. Fig 3g. The authors should include control groups infected with control siRNA.
5. Fig. 3f and 4a. The authors should show the interaction occurs at mitochondria.
6. In FigS1h, p62 was not altered, while LC3-II increased in Fig4c, d. The authors should show the autophagic flux to conclude the effect of MCL-1 on autophagy.
7. In fig 6a, the authors should examine the mitophagy level and damaged mitochondria quantitatively.
8. The authors should discuss the molecular mechanism underlying amyloid degradation by UMI-77.
9. The quantitative analyses should be performed in all Western blots.
10. The knockdown efficiency has to be shown in experiments using siRNA.

11. The authors should show the representative images in 2b, 3g and 5a.
12. In fig. 6a, the authors should show ratiometric images but not merged images.

Reviewers' comments:

Reviewer #1 (Remarks to the Author)

In the research article 'Targeting MCL-1 promotes mitophagy and ameliorates Alzheimer's disease pathologies', the authors show two novel findings, including 1) MCL-1 is a mitophagy receptor and directly binds to LC3A and interacts with ATG5, and 2) UMI-77 can binds on MCL-1 and UMI-77 ameliorates cognitive decline and amyloid pathologies in the APP/PS1 mouse model.

The authors had very detailed and series biochemistry, cell biology-based data and mouse experiments and with impressive amount of data. There are a couple of major concerns from this reviewer that the authors should consider:

1. Whether mitophagy plays a role in UMI-77-induced inhibition of AD memory loss and pathology in mice? It is convincing from the data that UMI-77 can induce mitophagy, and UMI-77 can forestall AD pathology. However, the current data do not provide direct evidence that UMI-77- forestalls AD through mitophagy. Although mitophagy induction can inhibits AD, it does not exclude possibilities that UMI-77-based non-autophagy pathway(s) may also inhibit AD. Some suggested experiments include to check whether overexpression of MCL-1 inhibits memory loss and pathology in the APP/PS1 mice. It may be too long to generate an MCL-1 overexpressed mouse strain and cross with the APP/PS1 mice; but it is capable that the authors to use AAV to overexpress Mcl-1 in hippocampus or the entorhinal cortex of the APP/PS1 mice, and to follow up behavioral and pathological studies.

Response:

We thank the reviewer for their appreciation of our study and the constructive comments. We have now addressed the comments in detail below. We have also performed new experiments and added a total of 22 new experimental panels to the manuscript. The text, newly added to the manuscript, is indicated in red. We hope that the reviewer finds our responses satisfactory. Please note that the figure citations in our responses below refer to the new (post-revision) figures.

As per reviewer's suggestion we have now overexpressed MCL-1 in hippocampus of six-month-old APP/PS1 mice using an AAV viral delivery system. We found that the overexpression of MCL-1 reduces extracellular amyloid- β plaque levels in the hippocampus and ameliorates cognitive decline in this model of Alzheimer's disease (Supplementary Fig. 10 and below). Interestingly, overexpression of MCL-1 also improved the learning and memory of wild-type mice (Supplementary Fig. 10a and below). This data indicates that MCL-1-mediated mitophagy plays an important role in neuronal function.

Overexpression of MCL-1 ameliorates cognitive decline in the APP/PS1 mouse model of Alzheimer's disease.

Supplementary Fig. 10a. Six-month-old C57BL/6 and APP/PS1 mice were overexpressed MCL-1 in hippocampus for one month. Latency to escape to a hidden platform in the Morris water maze during a 4-day training period ((WT, n=6), (WT (Ctrl vector), n=5), (WT (MCL-1 overexpression), n=6), (APP/PS1, n=6), (APP/PS1 (Ctrl vector), n=6), (APP/PS1 (MCL-1 overexpression), n=5). * p<0.05, t-test)). Source data are provided as a Supplementary Source Data file.

Supplementary Fig. 10b. Mice were treated as in Supplementary Fig. 10a and immunohistochemistry of hippocampus was performed to stain for amyloid plaques (6E10 antibody, green) and nuclei (DAPI, blue). Scale bar, 1000µm. Source data are provided as a Supplementary Source Data file.

Supplementary Fig. 10c. Mice were treated as in Supplementary Fig. 10a and brain lysates were

immunoblotted for MCL-1 and Tubulin. Source data are provided as a Supplementary Source Data file.

2. Can UMI-77 also induce macro-autophagy at the dose it induces mitophagy? (E.g., the authors mentioned 'sub-lethal doses, UMI-77 potently induces mitophagy'). Since 'UMI-77-induced mitophagy is mediated via the ATG5 autophagy pathway', if it interacts with the general autophagy protein ATG5, then UMI-77 will very likely to induce macroautophagy. If UMI-77 induces both mitophagy and macroautophagy, it is fine; the authors just need to provide the data.

Response:

We thank the reviewer for this comment. Our data indicates that UMI-77 does not induce macroautophagy, as indicated by lack of induction of degradation of p62 and ER marker Calnexin (see Fig 1g, 2d, 4c, S1g, S1i shown below). Additionally, UMI-77 did not significantly decrease the levels of GM130 (Golgi marker), Actin and GAPDH, but decreased the levels of the mitochondrial marker Tim23 (see Supplementary Fig. 1d shown below). On the other hand, we found that UMI-77 treatment can induce mitophagy as judged by reduction of Tim23 levels and increase in the mt-Keima signal, even in the absence of macroautophagy due to Beclin1 knockdown (see Supplementary Fig. 9b, 9c shown below). These data suggest that UMI-77 induces mitophagy, but not macroautophagy and that it can induce mitophagy independent of macroautophagy.

UMI-77 does not induce the degradation of p62 and ER marker Calnexin.

Figure 1g. HEK293T and HeLa cells were treated with 5 μ M UMI-77 for the indicated times and cell lysates were immunoblotted with indicated antibodies. Mitochondrial markers (Tom20, Tim23), endoplasmic reticulum marker (Calnexin), and cytosolic marker (Tubulin) were employed. The numbers under the blots represent the gray scale quantification (Tom20/Tubulin, Tim23/Tubulin). Source data are provided as a Figure 1 Source Data file.

Figure 2d. MCL-1-expressing HEK293T-MF2 cells were treated with 1 µg/mL doxycycline for the indicated times, cell lysates were immunoblotted with indicated antibodies. The numbers under the blot represent the gray scale quantification (Tim23/Tubulin). Source data are provided as a Figure 2 Source Data file.

Figure 4c. HeLa WT and quadruple KO (NBR1, TAXBP1, p62, and NDP52 knockout) cells were treated with 5 µM UMI-77 for the indicated times and cell lysates were immunoblotted with indicated antibodies. The numbers under the blots represent the gray scale quantification (Cox II/Tubulin, Tim23/Tubulin). Source data are provided as a Figure 4 Source Data file.

Supplementary Fig. 1g. U2OS and SH-SY5Y cells were treated with 5 µM UMI-77 for the indicated times and cell lysates were immunoblotted with indicated antibodies. The numbers under the blots represent the gray scale quantification (Tom20/Tubulin, Tim23/Tubulin). Source data are provided as a Supplementary Source Data file.

Supplementary Fig. 1i. HEK293T cells were treated with 5 μ M UMI-77 for the indicated times, and cell lysates were immunoblotted with indicated antibodies. The numbers under the blots represent the gray scale quantification (Tim23/Tubulin, p62/Tubulin). Source data are provided as a Supplementary Source Data file.

UMI-77 did not significantly decrease the levels of GM130 (Golgi marker), Actin and GAPDH, but decreased the levels of the mitochondrial marker Tim23.

Supplementary Fig. 1d. HEK293T cells were treated with 5 μ M UMI-77 for the indicated times and cell lysates were immunoblotted with indicated antibodies. The numbers under the blots represent the gray scale quantification (Tim23/Tubulin). Source data are provided as a Supplementary Source Data file.

UMI-77 induces mitophagy independent of Beclin1.

Supplementary Fig. 9b. HEK293T cells were transfected with Beclin1 siRNA for 48 h and treated with 5 μ M UMI-77 for 12 h. Cell lysates were immunoblotted for mitochondrial marker proteins (Tim23). The numbers under the blots represent the gray scale quantification (Tim23/Tubulin). siNC: scrambled siRNA. Source data are provided as a Supplementary Source Data file.

Supplementary Fig. 9c. Beclin1 knockdown HEK293T-mt-Keima cells were treated with 5 μ M UMI-77 for 12 h. The mitophagy levels were analyzed using t-test (data represent mean \pm S.E.M.; n=11. ** p<0.01, *** p<0.001.). siNC: scrambled siRNA. Source data are provided as a Supplementary Source Data file.

3. Mitophagy inhibition vs. mitophagy induction. This reviewer understands that MCL-1 induces mitophagy via LC3A in the condition of oxygen-glucose deprivation, while MCL-1 also inhibits mitophagy via AMBRA1 inhibition in the condition of MMP deprivation. But the molecular mechanisms on how MCL-1 balances mitophagy induction and mitophagy inhibition are still not clear.

Response:

We thank the reviewer for this comment. We also noticed that MCL-1 is reported to act as a suppressor of AMBRA1 to suppress mitophagy, under the conditions of FCCP or CCCP treatment (PMID: 31434979). It is reported that AMBRA1-mediated mitophagy is regulated by HUWE1, an E3 ubiquitin ligase, which could control mitochondrial protein ubiquitylation in cooperation with AMBRA1

during the mitophagy process in a PARKIN-free cellular system, in the condition of MMP deprivation (PMID: 30217973). And MCL-1 overexpression is sufficient to inhibit recruitment to mitochondria of HUWE1, upon AMBRA1-mediated mitophagy induction (PMID: 31434979). However, PINK1/PARKIN, but not HUWE1/AMBRA1, is the main pathway regulating mitophagy in cells with PARKIN, following MMP deprivation, during which the inhibition of MCL-1 on HUWE1 translocation to mitochondria may not inhibit mitophagy. What's more, these previous findings suggest that MCL-1 may inhibit ubiquitin-dependent mitophagy through inhibition of E3 ligase translocation to depolarized mitochondria, while our findings suggest that MCL-1-mediated mitophagy may be in an ubiquitin-independent way. In this way, our finding does not contradict the conclusion of this article (PMID: 31434979). We have addressed this point in the Discussion section of the manuscript. We hope to address this in our future studies.

Minor comment

1. 'However, no safe and effective mitophagy activators are available', 'Though mitophagy is suggested as a potential therapeutic approach for AD, there is a lack of safe and effective pharmacological compound that can cross blood-brain barrier and selectively induce mitophagy'. These statements are not correct. The NAD⁺ precursor nicotinamide riboside (NR) induces neuronal mitophagy and is safe based on the data from over 5 clinical trials (PMID: 31577933). And another anti-AD mitophagy inducer urolithin A is safe based on the evidence in a recent clinical trial (<https://www.nature.com/articles/s42255-019-0073-4>). Such studies should be mentioned in the discussion session.

Response:

We thank the reviewer for this pointing out these missing references. We have cited and discussed them in our revised manuscript and marked in red.

“Nicotinamide riboside (NR) is a safe and effective activator of neuronal mitophagy⁴⁷. NR is a precursor of NAD⁺ and can be metabolized to produce NAD⁺ in cells. NR reduces A β levels in APP/PS1 mice and has been tested in clinical trials^{47, 48}. Urolithin A (UA) is a natural, dietary, microflora-derived metabolite, which can induce mitophagy and ameliorate cognitive decline in the APP/PS1 mouse model^{49, 50}. However, the mechanism of mitophagy induced by these two drugs is not clear, highlighting the necessity to identify safe, effective, and clear mechanisms for mitophagy inducers that can become effective therapeutic approaches for the treatment of Alzheimer's disease.”

47. Lautrup, S., Sinclair, D.A., Mattson, M.P. & Fang, E.F. NAD(+) in Brain Aging and Neurodegenerative Disorders. *Cell Metab* **30**, 630-655 (2019).

48. Sorrentino, V. *et al.* Enhancing mitochondrial proteostasis reduces amyloid-beta proteotoxicity. *Nature* **552**, 187-+ (2017).

49. Andreux, P.A. *et al.* The mitophagy activator urolithin A is safe and induces a molecular signature of improved mitochondrial and cellular health in humans. *Nat Metab* **1**, 595-603 (2019).

50. Gong, Z. *et al.* Urolithin A attenuates memory impairment and neuroinflammation in APP/PS1 mice. *J Neuroinflammation* **16**, 62 (2019).

Reviewer #2 (Remarks to the Author):

The authors screened a library of FDA-approved drugs or drug candidates, and identified UMI-77, a previously reported BH3-mimetic to induce apoptosis, as a novel mitophagy activator at sublethal doses. They showed that MCL-1 is LIR motif to directly bind to LC3 and to mediate mitophagy in response to oxygen-glucose deprivation in an ATG-5 dependent manner. The authors also claimed that UMI-77 can induce mitophagy in vivo and ameliorate the AD phenotypes of APP/PS1 mice. The authors provide interesting set of data to show UMI-77 activates Mcl-1 dependent mitophagy. However, they do not show how MCL-1 is activated by UMI-77 to induce mitophagy. Does its inhibition on the interaction between MCL-1 and Bax/Bak lead to the exposure LIR of MCL1 for mitophagy? It would be important to dissect the interplay between apoptosis and mitophagy as the LIR motif is within the BH1 domain of Mcl-1. Also, strong evidence and clearer images are needed to show the mitophagy induction ability in vivo (in brain) by UMI-77.

1.Does its inhibition on the interaction between MCL-1 and Bax/Bak lead to the exposure LIR of MCL1 for mitophagy?

Response:

We thank the reviewer for their appreciation of our study and the constructive comments. We have now addressed the comments in detail below. We have also performed new experiments and added a total of 22 new experimental panels to the manuscript. The text, newly added to the manuscript, is indicated in red. We hope that the reviewer finds our responses satisfactory. Please note that the figure citations in our responses below refer to the new (post-revision) figures.

Given that LIR motif is within the BH1 domain of MCL-1, we hypothesized that the UMI-77-induced release of MCL-1 from Bax/Bak allows MCL-1 to interact with LC3, thereby promoting mitophagy. MCL-1 has been shown to interact with Bax through BH3 domain (PMID: 19968986). We generated L213A/D218A mutant of MCL-1 (BH3 domain mutant) and confirmed that it reduces the interaction between MCL-1 and Bax. Overexpression of both MCL-1-WT and MCL-M (L213A/D218A) induce mitophagy. Overexpression of MCL-M (L213A/D218A) can enhance mitochondrial autophagy and result in further degradation of TIM23 at 48h (see Supplementary Fig. 5a and 5b shown below).

We also show that UMI-77 or OGD inhibits the interaction between MCL-1 and Bax/Bak (see Figure 3c and 5f shown below) and that OGD enhances the interaction between MCL-1 and LC3A (see Figure 5f shown below). This provides more evidence for our model. We hope that this addresses the reviewer's concern.

The L213A/D218A mutant of MCL-1 promotes mitophagy.

Supplementary Fig. 5a. HEK293T cells were transfected with MCL-1-WT-3xFlag or MCL-1-M-3xFlag (L213A/D218A) for 24 h, and the interaction between Flag-tagged MCL-1 and endogenous Bax was analyzed by immunoprecipitation. Source data are provided as a Supplementary Source Data file.

Supplementary Fig. 5b. HEK293T cells were transfected with pcDNA3.1-MCL-1-WT (wild-type) or pcDNA3.1-MCL-1-M (L213A/D218A) plasmid for the indicated times and cell lysates were immunoblotted with indicated antibodies. The numbers under the blots represent the gray scale quantification (Tim23/Tubulin). Source data are provided as a Supplementary Source Data file.

UMI-77 or OGD inhibits the interaction between MCL-1 and Bax/Bak and OGD enhances the interaction between MCL-1 and LC3A.

Figure 3c. HEK293T cells were co-transfected with MCL-1-3xFlag and HA-LC3A for 24 h, treated with UMI-77 (10 μ M) for 4 h, and the interaction of MCL-1 with HA-LC3A and endogenous Bax was analyzed by immunoprecipitation. Source data are provided as a Figure 3 Source Data file.

Figure 5f. HEK293T cells were co-transfected with MCL-1-3xFlag and HA-LC3A for 24 h, treated with OGD for 5 h, and the interaction of Flag-tagged MCL-1 with HA-LC3A and endogenous Bax was analyzed by immunoprecipitation. Source data are provided as a Figure 5 Source Data file.

2. It would be important to dissect the interplay between apoptosis and mitophagy.

Response: Please see the response to the Specific comments #1.

3. Strong evidence and clearer images are needed to show the mitophagy induction ability *in vivo* (in brain) by UMI-77.

Response: We have now repeated the experiment shown in Figure 6a and quantified the data to confirm induction of mitophagy *in vivo* (in brain) by UMI-77. We have also performed more *in vivo* experiments and the results are shown as followed (see the response to comment #6).

The Induction of mitophagy *in vivo* (in brain) by UMI-77.

Figure 6a. mt-Keima signal induction by UMI-77 in the mt-Keima mouse brain hippocampus

samples. Mice were injected with 10 mg/kg UMI-77 for 6 h and samples were analyzed by fluorescence microscopy (** $p < 0.01$, t-test). Scale bar, 100 μm . Source data are provided as a Figure 6 Source Data file.

Specific comments include:

1. In the paper, the cells were treated with UMI-77 for only 12 hours, and then the cell viability assay was performed. The treatment duration should be extended to at least 24 hours to determine if the drug is lethal at 5 μM . Also, Live/dead kit is used in the paper to stain dead cells with damaged cell membranes, which is a later apoptosis event. It is known that UMI-77, as low as 3 μM , can induce caspase 3 cleavage in pancreatic cancer cells. (PMID: 24019208).

Response:

Thank you for pointing this detail out. We have performed prolonged time course experiments, as suggested, and tested caspase-3 cleavage assays at the sub-lethal dose of 5 μM at the 24h time point. As indicated by our caspase-3 activity data (see Supplementary Fig. 2b, 2c, and 2d shown below), UMI-77 cannot induce apoptosis in HEK293T and SH-SY5Y cells at the sub-lethal dose of 5 μM at the 24h time point. This effect may be due to cell line differences used in the reference cited by the reviewer, which used pancreatic cancer cell lines, and our study (HEK293T and SH-SY5Y cells). This new data confirms that UMI-77 induces mitophagy independent of apoptosis.

UMI-77 induces mitophagy independent of apoptosis.

Supplementary Fig. 2b. HEK293T cells were transfected with pcDNA3.1-mt-Keima plasmid for 24 h and treated with UMI-77 (0 μM , 2 μM , 5 μM , 10 μM , 20 μM) with or without Z-VAD-fmk (50 μM) for 24 h. Apoptosis levels was estimated using Caspase-Glo[®] 3/7 Assay. One-way ANOVA (data represent mean \pm S.E.M.; $n=4$. **** $p < 0.0001$, * $p < 0.05$). Source data are provided as a Supplementary Source Data file.

Supplementary Fig. 2c. As in Supplementary Fig. 2b, except SH-SY5Y cells were used. Source data are provided as a Supplementary Source Data file.

Supplementary Fig. 2d. Cells were treated with staurosporine (0 μM, 5 μM, 10 μM) for 3 h. Apoptosis levels were estimated as in Supplementary Fig. 2b. One-way ANOVA (data represent mean ± S.E.M.; n=4. **** p<0.0001, * p<0.05). Source data are provided as a Supplementary Source Data file.

2. Other mitophagy receptors, such as BNIP3, FUNDC1, NIX, should be checked if they also are involved in UMI-77 induced mitophagy, as these receptors have been suggested to participate in hypoxia induced mitophagy.

Response: We thank the reviewer for the insightful comments and suggestions. We have now performed knockdown experiments for these three mitophagy receptors and tested the mitophagy levels. Our results indicate that none of the mitophagy receptors play role in the UMI-77-induced mitophagy (see Supplementary Fig. 8a-c shown below).

UMI-77 induces mitophagy independent of BNIP3, NIX, FUNDC1.

Supplementary Fig. 8a. HEK293T cells were transfected with NIX siRNA for 48 h and treated with 5 μ M UMI-77 for 12 h. Cell lysates were immunoblotted for mitochondrial marker proteins (Tom20, Tim23). The numbers under the blots represent the gray scale quantification (Tom20/Tubulin, Tim23/Tubulin). siNC: scrambled siRNA. Source data are provided as a Supplementary Source Data file.

Supplementary Fig. 8b. As in Supplementary Fig. 8a, except Bnip3 siRNA was used. Source data are provided as a Supplementary Source Data file.

Supplementary Fig. 8c. As in Supplementary Fig. 8a, except *FUNDC1* siRNA was used. Source data are provided as a Supplementary Source Data file.

Supplementary Fig. 8d. Cells were treated as in Supplementary Fig. 8c and total RNA were extracted by FastPure® Cell/Tissue Total RNA Isolation Kit for *FUNDC1* qPCR. Data were analyzed using one-way ANOVA (data represent mean \pm S.E.M.; $n=3$. **** $p<0.0001$). siNC: scrambled siRNA. Source data are provided as a Supplementary Source Data file.

3. In addition to mitochondria, MCL1 also localizes at the endoplasmic reticulum, the Golgi apparatus and the peroxisomes. Why does UMI-77 treatment only induce mitophagy?

Response:

We thank the reviewer for this comment. By searching The Human Protein Atlas and GeneCards databases, we found that MCL-1 was mainly distributed in mitochondria, while the truncated type of MCL-1 known at present was distributed in cytoplasm, and its role was mainly to block the interaction between MCL-1 and Bax on mitochondria and promote apoptosis (PMID: 10837489). Therefore, we

speculate that MCL-1 distributed in the cytoplasm or elsewhere has different functions from MCL-1 on mitochondria, may not have the interaction with Bax, and thus will not respond to UMI-77.

We thank the reviewer for this comment. Our data indicates that UMI-77 does not induce macroautophagy, as indicated by lack of induction of degradation of p62 and ER marker Calnexin (see Fig 1g, 2d, 4c, S1g, S1i shown below). Additionally, UMI-77 did not significantly decrease the levels of GM130 (Golgi marker), Actin and GAPDH, but decreased the levels of the mitochondrial marker Tim23 (see Supplementary Fig. 1d shown below). On the other hand, we found that UMI-77 treatment can induce mitophagy as judged by reduction of Tim23 levels and increase in the mt-Keima signal, even in the absence of macroautophagy due to Beclin1 knockdown (see Supplementary Fig. 9b, 9c shown below). These data suggest that UMI-77 induces mitophagy, but not macroautophagy and that it can induce mitophagy independent of macroautophagy.

UMI-77 does not induce the degradation of p62 and ER marker Calnexin.

Figure 1g. HEK293T and HeLa cells were treated with 5 μ M UMI-77 for the indicated times and cell lysates were immunoblotted with indicated antibodies. Mitochondrial markers (Tom20, Tim23), endoplasmic reticulum marker (Calnexin), and cytosolic marker (Tubulin) were employed. The numbers under the blots represent the gray scale quantification (Tom20/Tubulin, Tim23/Tubulin). Source data are provided as a Figure 1 Source Data file.

Figure 2d. MCL-1-expressing HEK293T-MF2 cells were treated with 1 μ g/mL doxycycline for the indicated times, cell lysates were immunoblotted with indicated antibodies. The numbers under the blots represent the gray scale quantification (Tim23/Tubulin). Source data are provided as a Figure 2 Source Data file.

Figure 4c. HeLa WT and quadruple KO (NBR1, TAXBP1, p62, and NDP52 knockout) cells were treated with 5 μ M UMI-77 for the indicated times and cell lysates were immunoblotted with indicated antibodies. The numbers under the blots represent the gray scale quantification (Cox II/Tubulin, Tim23/Tubulin). Source data are provided as a Figure 4 Source Data file.

Supplementary Fig. 1g. U2OS and SH-SY5Y cells were treated with 5 μ M UMI-77 for the indicated times and cell lysates were immunoblotted with indicated antibodies. The numbers under the blots represent the gray scale quantification (Tom20/Tubulin, Tim23/Tubulin). Source data are provided as a Supplementary Source Data file.

Supplementary Fig. 1i. HEK293T cells were treated with 5 μ M UMI-77 for the indicated times, and cell lysates were immunoblotted with indicated antibodies. The numbers under the blots represent the gray scale quantification (Tim23/Tubulin, p62/Tubulin). Source data are provided as a Supplementary Source Data file.

UMI-77 did not significantly decrease the levels of GM130 (Golgi marker), Actin and GAPDH, but decreased the levels of the mitochondrial marker Tim23.

Supplementary Fig. 1d. HEK293T cells were treated with 5 μ M UMI-77 for the indicated times and cell lysates were immunoblotted with indicated antibodies. The numbers under the blots represent the gray scale quantification (Tim23/Tubulin). Source data are provided as a Supplementary Source Data file.

UMI-77 induces mitophagy independent of Beclin1.

Supplementary Fig. 9b. HEK293T cells were transfected with Beclin1 siRNA for 48 h and treated with 5 μ M UMI-77 for 12 h. Cell lysates were immunoblotted for mitochondrial marker proteins (Tom20). The numbers under the blots represent the gray scale quantification (Tom20/Tubulin). siNC: scrambled siRNA. Source data are provided as a Supplementary Source Data file.

4. Images in Figure 5e are not convincing.

Response:

We have now repeated this experiment (see Figure 5e below). Lane #4 is expected to show a decrease in Tom20/Tim23 levels significantly, which does show a strong induction of mitophagy. All the other lanes display only a slight mitophagy induction, which indicates that LIR mutants of MCL-1 partially rescue the effect of OGD. The incompleteness of the effect may be due to the residual levels of MCL-1 present following the knockdown.

We also show that OGD enhances the interaction between MCL-1 and LC3A (see Figure 5f below), which further supports our model. We hope that the new data addresses the reviewer's concern.

OGD enhances the interaction between MCL-1 and LC3A

Figure 5f. HEK293T cells were co-transfected with MCL-1-3xFlag and HA-LC3A for 24 h, treated with OGD for 5 h, and the interaction of MCL-1 with LC3A and Bax was analyzed by immunoprecipitation. Source data are provided as a Figure 5 Source Data file.

5. Figure 4d: the LC3 bands in the right panel are weird. UMI-77 induces autophagy in ATG5 KO cells?

Response:

This is an interesting yet puzzling observation. UMI-77 induces accumulation of the unlipidated form of LC3 (LC3-I) following in ATG5 KO cells, however, this does not mean that it induces autophagy. The rationale behind this LC3-I accumulation is not clear. This phenomenon has been reported previously (PMID: 26476415). It is a consequence of the inability of ATG5 KO cells to proceed with autophagy, despite autophagy initiation. The key finding in this experiment is that UMI-77-induced mitophagy (i.e., Tim23 degradation) is blocked by ATG5 KO, indicating that the UMI-77-induced mitophagy is ATG5-dependent. Thus, the UMI-77-induced LC3-I accumulation in ATG5 KO cells is not unexpected, as these cells fail to proceed with UMI-77-induced mitophagy, despite its initiation.

6. Figure 6f: the mice number of APP/PS1 (veh) set is 5 when the level of IL-10 was measured, while the number is 4 when TNF-alpha and IL-6 were measured. Please explain

Response:

We thank the reviewer for this comment. A sample was lost during the experiment. We have now performed additional *in vivo* experiments:

We have now overexpressed MCL-1 in hippocampus of six-month-old APP/PS1 mice using AAV viral delivery system. We found that the overexpression of MCL-1 reduces extracellular amyloid- β plaque levels in the hippocampus and ameliorates cognitive decline in this model of Alzheimer's disease (Supplementary Fig. 10a-c and below). Interestingly, overexpression of MCL-1 also improved the learning and memory of wild-type mice (Supplementary Fig. 10a and below). This indeed indicates that MCL-1-mediated mitophagy plays an important role in neuronal function.

Overexpression of MCL-1 ameliorates cognitive decline in the APP/PS1 mouse model of Alzheimer's disease.

Supplementary Fig. 10a. Six-month-old C57BL/6 and APP/PS1 mice were overexpressed MCL-1 in hippocampus for one month. Latency to escape to a hidden platform in the Morris water maze during a 4-day training period ((WT, n=6), (WT (Ctrl), n=5), (WT (MCL-1), n=6), (APP/PS1, n=6), (APP/PS1 (Ctrl), n=6), (APP/PS1 (MCL-1), n=5). * $p < 0.05$, t-test)). Source data are provided as a Supplementary Source Data file.

Supplementary Fig. 10b. Mice were treated as in Supplementary Fig. 10a and IHC of hippocampus was performed to stain for amyloid plaques (6E10 antibody, green) and nuclei (DAPI, blue). Scale bar, 1000 μ m. Source data are provided as a Supplementary Source Data file.

Supplementary Fig. 10c. Mice were treated as in Supplementary Fig. 10a and brain lysates were immunoblotted for MCL-1 and Tubulin. Source data are provided as a Supplementary Source Data file.

Reviewer #3 (Remarks to the Author):

In this study, the authors identified UMI-77 as a novel mitophagy inducer by screening FDA-approved drugs, and UMI-77-induced mitophagy is mediated by MCL-1. Besides, the mitophagy induction by UMI-77 improved the pathogenic phenotypes in model mice for Alzheimer's disease.

This study is novel and interesting. However, the authors need to perform experiments to confirm their conclusion.

1. The authors suggest that the release of MCL-1 from Bax/Bak allows MCL-1 to interact with LC3, thereby promoting mitophagy. To confirm this, the authors should show the mitophagic activity of MCL-1 mutant, which cannot interact with Bax/Bak. Also, the authors should show that UMI-77 or OGD inhibits the interaction between MCL-1 and Bax/Bak. In addition, the authors should show OGD enhances the interaction between MCL-1 and LC3A.

Response:

We thank the reviewer for their appreciation of our study and the constructive comments. We have now addressed the comments in detail below. We have also performed new experiments and added a total of 22 new experimental panels to the manuscript. The text, newly added to the manuscript, is indicated in red. We hope that the reviewer finds our responses satisfactory. Please note that the figure citations in our responses below refer to the new (post-revision) figures.

We thank the reviewer the constructive comments. Given LIR motif is within the BH1 domain of MCL-1, we hypothesized that the UMI-77-induced release of MCL-1 from Bax/Bak allows MCL-1 to interact with LC3, thereby promoting mitophagy. MCL-1 has been shown to interact with Bax through BH3 domain (PMID: 19968986). We generated L213A/D218A mutant of MCL-1 (BH3 domain mutant) and confirmed that it reduces the interaction between MCL-1 and Bax. Overexpression of both MCL-1-WT and MCL-M (L213A/D218A) induce mitophagy. Overexpression of MCL-M (L213A/D218A) can enhance mitochondrial autophagy and result in further degradation of TIM23 at 48h (see Supplementary Fig. 5a and 5b shown below).

Moreover, we also show that UMI-77 or OGD inhibits the interaction between MCL-1 and Bax/Bak (see Figure 3c and 5f shown below) and that OGD enhances the interaction between MCL-1 and LC3A (see Figure 5f shown below). This provides more evidence for our model. We hope that these new lines of evidence address the reviewer's concern.

The L213A/D218A mutant of MCL-1 promotes mitophagy.

Supplementary Fig. 5a. HEK293T cells were transfected with MCL-1-WT-3xFlag or MCL-1-M-3xFlag (L213A/D218A) for 24 h, and the interaction between MCL-1 and Bax was analyzed by immunoprecipitation. Source data are provided as a Supplementary Source Data file.

Supplementary Fig. 5b. HEK293T cells were transfected with pcDNA3.1-MCL-1-WT (wild-type) or pcDNA3.1-MCL-1-M (L213A/D218A) plasmid for the indicated times and cell lysates were immunoblotted with indicated antibodies. The numbers under the blots represent the gray scale quantification (Tim23/Tubulin). Source data are provided as a Supplementary Source Data file.

UMI-77 or OGD inhibits the interaction between MCL-1 and Bax/Bak and OGD enhances the interaction between MCL-1 and LC3A.

Figure 3c. HEK293T cells were co-transfected with MCL-1-3xFlag and HA-LC3A for 24 h, treated with UMI-77 (10 μ M) for 4 h, and the interaction of MCL-1 with LC3A and Bax was analyzed by

immunoprecipitation. Source data are provided as a Figure 3 Source Data file.

Figure 5f. HEK293T cells were co-transfected with MCL-1-3xFlag and HA-LC3A for 24 h, treated with OGD for 5 h, and the interaction of MCL-1 with LC3A and Bax was analyzed by immunoprecipitation. Source data are provided as a Figure 5 Source Data file.

2. MCL-1 is reported to inhibit autophagy mediated through Beclin-1 interaction. Is MCL-1-induced mitophagy Beclin-1-dependent?

Response:

We thank the reviewer an interesting suggestion. Given that overexpression of MCL-1 can inhibit autophagy through its interaction with Beclin1 (PMID: 21139567), we think overexpression of MCL-1 could not release and activate Beclin1, suggesting that Beclin1 was not related to MCL-1-induced mitophagy. We also show that UMI-77 induce mitophagy is independent of Beclin-1 (see Supplementary Fig. 9b and 9c below).

UMI-77 induces mitophagy independent of Beclin1.

Supplementary Fig. 9b. HEK293T cells were transfected with Beclin1 siRNA for 48 h and treated with 5 μ M UMI-77 for 12 h. Cell lysates were immunoblotted for mitochondrial marker proteins (Tim23). The numbers under the blots represent the gray scale quantification (Tim23/Tubulin). siNC: scrambled siRNA. Source data are provided as a Supplementary Source Data file.

Supplementary Fig. 9c. Beclin1 knockdown HEK293T-mt-Keima cells were treated with 5 μ M UMI-77 for 12 h. The mitophagy levels were analyzed using t-test (data represent mean \pm S.E.M.; n=11. ** p<0.01, *** p<0.001.). The siRNA knockdown efficiency was shown using western blotting. siNC: scrambled siRNA. Source data are provided as a Supplementary Source Data file.

3. Fig2c-e show overexpression of MCL-1 induced mitophagy in the absence of UMI-77. However, in Fig 3c, overexpression of MCL-1 did not induce LC3A-MCL-1 interaction in the absence of UMI-77. In Fig 3g, the mitophagy levels were not different between WT and mutants in the absence of UMI-77. These indicate overexpression of MCL-1 does not induce mitophagy in the absence of UMI-77. How do the authors reconcile the apparent discrepancy?

Response:

We thank the reviewer for this comment. We think this may be due to a western blotting exposure problem (Fig. 3c) and the way of data presented (Fig. 3g, where UMI-77 untreated group is used as the control group). We have now repeated these two experiments and confirmed that overexpression of MCL-1 indeed induces interaction between LC3A and MCL-1 in the absence of UMI-77 (see Figure 3c shown below). Moreover, compared to wild-type MCL-1, the mutants reduced the UMI-77-induced mitophagy as well as basal mitophagy (see Figure 3g shown below). We hope that this addresses the reviewer's concern.

Overexpression of MCL-1 indeed induces interaction between LC3A and MCL-1 in the absence of UMI-77.

Figure 3c. HEK293T cells were co-transfected with MCL-1-3xFlag and HA-LC3A for 24 h, treated with UMI-77 (10 μ M) for 4 h, and the interaction of MCL-1 with LC3A and Bax was analyzed by immunoprecipitation. Source data are provided as a Figure 3 Source Data file.

The mutants of MCL-1 reduced the UMI-77-induced mitophagy as well as basal mitophagy, comparing to wild-type MCL-1.

Figure 3g. HEK293T-MCL-1-knockdown cells were co-transfected with MCL-1 WT or indicated mutants and mt-Keima plasmid for 48 h, treated with UMI-77 (5 μM) for 12 h. The mitophagy levels were quantified by t-test (data represent mean ± S.E.M.; n=3, **** p<0.0001, ** p<0.01, ns, not significant.). Source data are provided as a Figure 3 Source Data file.

4. Fig 3g. The authors should include control groups infected with control siRNA.

Response:

We thank the reviewer for the suggestion. We have now re-generated an MCL-1 knockdown stable cell line and used it in the experiments shown in Fig. 3g (The knockdown efficiency is shown in supplementary Fig. 3c and below). We hope that this addresses the reviewer's concern.

The knockdown efficiency of MCL-1 knockdown stable cell line used in the experiments.

Supplementary Fig. 3c. HEK293T-MCL-1-knockdown (HEK293T-shMCL-1) and HEK293T-control-knockdown cells (HEK293T-shNC) were treated with UMI-77 as Figure 3g shown and cell lysates were immunoblotted with indicated antibodies. Source data are provided as a Supplementary Source Data file.

5. Fig. 3f and 4a. The authors should show the interaction occurs at mitochondria.

Response:

We thank the reviewer for the suggestion. We have now performed higher magnification analyses of these immunofluorescence experiments and show co-localization of the PLA dots on mitochondria using Mitotracker. As shown in Supplementary Fig. 7 and below, PLA dots co-localized with mitochondria in HeLa and HeLa quadruple KO, suggesting that the interaction occurs at mitochondria.

MCL-1 interacts with LC3A on mitochondria.

Supplementary Fig. 7. Wild-type and quadruple KO (NBR1, TAXBP1, p62, and NDP52 knockout) HeLa cells were treated with UMI-77 (5 μ M) for 3h and a PLA assay for MCL-1 and LC3A was performed. Cells were counter-stained with MitoTracker™ Deep Red FM. Scale bar, 5 μ m. Source data are provided as a Supplementary Source Data file.

6. In FigS1h, p62 was not altered, while LC3-II increased in Fig4c, d. The authors should show the autophagic flux to conclude the effect of MCL-1 on autophagy.

Response:

We thank the reviewer for the suggestion. We have now performed an autophagy flux assay. E64D is a lysosomal inhibitor that can be used in autophagy flux assay (PMID: 18188003). As E64D was

able to increase the level of LC3-II, UMI-77 may induce the increase of the autophagy flux (see Supplementary Fig. 1c shown below).

Our data indicates that UMI-77 does not induce macroautophagy, as indicated by lack of induction of degradation of p62 and ER marker Calnexin (see Fig 1g, 2d, 4c, S1g, S1i shown below). Additionally, UMI-77 did not significantly decrease the levels of GM130 (Golgi marker), Actin and GAPDH, but decreased the levels of the mitochondrial marker Tim23 (see Supplementary Fig. 1d shown below). On the other hand, we found that UMI-77 treatment can induce mitophagy as judged by reduction of Tim23 levels and increase in the mt-Keima signal, even in the absence of macroautophagy due to Beclin1 knockdown (see Supplementary Fig. 9b, 9c shown below). These data suggest that UMI-77 induces mitophagy, but not macroautophagy and that it can induce mitophagy independent of macroautophagy.

UMI-77 induces the increase of the autophagy flux.

Supplementary Fig. 1c. HEK293T cells were treated with UMI-77 and E64D, cell lysates were immunoblotted with indicated antibodies. The numbers under the blots represent the gray scale quantification (LC3II/Tubulin). Source data are provided as a Supplementary Source Data file.

UMI-77 does not induce the degradation of p62 and ER marker Calnexin.

Figure 1g. HEK293T and HeLa cells were treated with 5 μM UMI-77 for the indicated times and cell lysates were immunoblotted with indicated antibodies. Mitochondrial markers (Tom20, Tim23), endoplasmic reticulum marker (Calnexin), and cytosolic marker (Tubulin) were employed. The numbers under the blots represent the gray scale quantification (Tom20/Tubulin, Tim23/Tubulin). Source data are provided as a Figure 1 Source Data file.

Figure 2d. MCL-1-expressing HEK293T-MF2 cells were treated with 1 µg/mL doxycycline for the indicated times, cell lysates were immunoblotted with indicated antibodies. The numbers under the blots represent the gray scale quantification (Tim23/Tubulin). Source data are provided as a Figure 2 Source Data file.

Figure 4c. HeLa WT and quadruple KO (NBR1, TAXBP1, p62, and NDP52 knockout) cells were treated with 5 µM UMI-77 for the indicated times and cell lysates were immunoblotted with indicated antibodies. The numbers under the blots represent the gray scale quantification (Cox II/Tubulin, Tim23/Tubulin). Source data are provided as a Figure 4 Source Data file.

Supplementary Fig. 1g. U2OS and SH-SY5Y cells were treated with 5 µM UMI-77 for the indicated times and cell lysates were immunoblotted with indicated antibodies. The numbers under the blots represent the gray scale quantification (Tom20/Tubulin, Tim23/Tubulin). Source data are provided as a Supplementary Source Data file.

Supplementary Fig. 1i. HEK293T cells were treated with 5 μ M UMI-77 for the indicated times, and cell lysates were immunoblotted with indicated antibodies. The numbers under the blots represent the gray scale quantification (Tim23/Tubulin, p62/Tubulin). Source data are provided as a Supplementary Source Data file.

UMI-77 did not significantly decrease the levels of GM130 (Golgi marker), Actin and GAPDH, but decreased the levels of the mitochondrial marker Tim23.

Supplementary Fig. 1d. HEK293T cells were treated with 5 μ M UMI-77 for the indicated times and cell lysates were immunoblotted with indicated antibodies. The numbers under the blots represent the gray scale quantification (Tim23/Tubulin). Source data are provided as a Supplementary Source Data file.

UMI-77 induces mitophagy independent of Beclin1.

Supplementary Fig. 9b. HEK293T cells were transfected with Beclin1 siRNA for 48 h and treated with 5 μ M UMI-77 for 12 h. Cell lysates were immunoblotted for mitochondrial marker proteins (Tim23). The numbers under the blots represent the gray scale quantification (Tim23/Tubulin). siNC: scrambled siRNA. Source data are provided as a Supplementary Source Data file.

Supplementary Fig. 9c. Beclin1 knockdown HEK293T-mt-Keima cells were treated with 5 μ M UMI-77 for 12 h. The mitophagy levels were analyzed using t-test (data represent mean \pm S.E.M.; n=11. ** p<0.01, *** p<0.001.). siNC: scrambled siRNA. Source data are provided as a Supplementary Source Data file.

7. In fig 6a, the authors should examine the mitophagy level and damaged mitochondria quantitatively.

Response:

We thank the reviewer for the suggestion. We have now repeated this experiment and quantified the data (see Fig 6a shown below).

The Induction of mitophagy *in vivo* (in brain) by UMI-77.

Figure 6a. mt-Keima signal induction by UMI-77 in the mt-Keima mouse brain hippocampus samples. Mice were injected with 10 mg/kg UMI-77 for 6 h and samples were analyzed by fluorescence microscopy (** p<0.01, t-test). Scale bars, 100 μ m. Source data are provided as a Figure 6 Source Data file.

8. The authors should discuss the molecular mechanism underlying amyloid degradation by UMI-77.

Response:

We thank the reviewer for the comment. We have now addressed this in the Discussion section (see the text marked in red in the manuscript and given below).

“Risk factors in AD patients, such as genetic mutation, aging, and environmental factors, promote the production of reactive oxygen species (ROS) and lead to mitochondrial dysfunction, as well as the production of A β plaques. The abnormal mitochondrial function accelerates the production of ROS and fueling this process. Therefore, degradation of damaged mitochondria by mitophagy can prevent excessive production of ROS, reducing the production of A β production⁵¹. On the other hand, A β can be transported into cells and accumulated in mitochondria, and this interaction inhibits mitochondrial function, elevates ROS levels, and alters mitochondrial dynamics⁵²⁻⁵⁴. We report that UMI-77-induced mitophagy effectively reduces A β plaque levels, which may be achieved through decreased production of ROS and the A β deposited in the damaged mitochondria. Our work strongly supports UMI-77 as a putative drug for the treatment of Alzheimer's disease.”

51. Lin, Q. *et al.* PINK1-parkin pathway of mitophagy protects against contrast-induced acute kidney injury via decreasing mitochondrial ROS and NLRP3 inflammasome activation. *Redox Biol* **26**, 101254 (2019).
52. Hansson Petersen, C.A. *et al.* The amyloid beta-peptide is imported into mitochondria via the TOM import machinery and localized to mitochondrial cristae. *Proc Natl Acad Sci U S A* **105**, 13145-13150 (2008).

53. Pagani, L. & Eckert, A. Amyloid-Beta interaction with mitochondria. *Int J Alzheimers Dis* **2011**, 925050 (2011).
54. Tillement, L., Lecanu, L. & Papadopoulos, V. Alzheimer's disease: effects of beta-amyloid on mitochondria. *Mitochondrion* **11**, 13-21 (2011).

9. The quantitative analyses should be performed in all Western blots.

Response:

We thank the reviewer for the comment. We have now performed the suggested quantifications (see Fig. 1g, 1h, 2a, 2d, 3h, 4c, 4d, 5b, 5e. Supplementary Fig. 1c, 1d, 1e, 1g, 1h, 1i, 5b, 8a, 8b, 8c, 9b given below).

Quantifications of all western blots.

Figure 1g. HEK293T and HeLa cells were treated with 5 μ M UMI-77 for the indicated times and cell lysates were immunoblotted with indicated antibodies. Mitochondrial markers (Tom20, Tim23), endoplasmic reticulum marker (Calnexin), and cytosolic marker (Tubulin) were employed. The numbers under the blots represent the gray scale quantification (Tom20/Tubulin, Tim23/Tubulin). Source data are provided as a Figure 1 Source Data file.

Figure 1h. HEK293T and HeLa cells were treated with 5 μ M UMI-77 in the presence or absence of MG-132, E64D and NH₄Cl/Leupeptin for 12 h, and the mitochondrial marker proteins (Tom20, Tim23) were detected by western blotting. The numbers under the blots represent the gray scale quantification (Tom20/Tubulin, Tim23/Tubulin). Source data are provided as a Figure 1 Source Data file.

Figure 2a. HEK293T cells were transfected with MCL-1 shRNA for 60 h and treated with 5 μ M UMI-77 for 12 h. Cell lysates were immunoblotted for mitochondrial marker proteins (Tom20, Tim23). The numbers under the blots represent the gray scale quantification (Tom20/Tubulin, Tim23/Tubulin). shNC: scrambled shRNA. Source data are provided as a Figure 2 Source Data file.

Figure 2d. MCL-1-expressing HEK293T-MF2 cells were treated with 1 μ g/mL doxycycline for the indicated times, cell lysates were immunoblotted with indicated antibodies. The numbers under the blots represent the gray scale quantification (Tim23/Tubulin). Source data are provided as a Figure 2 Source Data file.

Figure 3h. SH-SY5Y MCL-1-knockdown cell line was transfected with expression plasmids encoding MCL-1 WT or indicated mutants for 24 h, treated with UMI-77 (5 μ M) for 12 h. Cell lysates were immunoblotted with indicated antibodies. The numbers under the blots represent the gray scale quantification (Tim23/Tubulin). Source data are provided as a Figure 3 Source Data file.

Figure 4c. HeLa WT and quadruple KO (NBR1, TAXBP1, p62, and NDP52 knockout) cells were treated with 5 μ M UMI-77 for the indicated times and cell lysates were immunoblotted with indicated antibodies. The numbers under the blots represent the gray scale quantification (Cox II/Tubulin, Tim23/Tubulin). Source data are provided as a Figure 4 Source Data file.

Figure 4d. MEF WT and *atg5* knockout cells were treated with 5 μ M UMI-77 for the indicated times, and mitochondrial marker protein Tim23 and LC3 were detected by western blotting. The numbers under the blots represent the gray scale quantification (Tim23/Tubulin). Source data are provided as a Figure 4 Source Data file.

Figure 5b. HEK293T-mt-Keima cells were infected with lentiviral particles encoding MCL-1 shRNA for 24 h and treated with OGD for 5 h. Cell lysates were immunoblotted with indicated antibodies. The numbers under the blots represent the gray scale quantification (Cox II/Tubulin, Tim23/Tubulin). shNC: scrambled shRNA. Source data are provided as a Figure 5 Source Data file.

Figure 5e. HEK293T *MCL-1*-knockdown cells were transfected with *MCL-1* WT or indicated mutants for 24 h, treated with OGD for 5 h. Cell lysates were immunoblotted with indicated antibodies. The numbers under the blots represent the gray scale quantification (Tom20/Tubulin, Tim23/Tubulin). Source data are provided as a Figure 5 Source Data file.

Supplementary Fig. 1c. HEK293T cells were treated with UMI-77 and E64D, cell lysates were immunoblotted with indicated antibodies. The numbers under the blots represent the gray scale quantification of the upper band (LC3 II/Tubulin). Source data are provided as a Supplementary Source Data file.

Supplementary Fig. 1d. HEK293T cells were treated with 5 μ M UMI-77 for the indicated times and cell lysates were immunoblotted with indicated antibodies. The numbers under the blots represent the gray scale quantification (Tim23/Tubulin). Source data are provided as a Supplementary Source Data file.

Supplementary Fig. 1e. U2OS cells were treated with 1 μ M staurosporine (STS) for the indicated

times, cell lysates were immunoblotted with indicated antibodies. The numbers under the blots represent the gray scale quantification (Tom20/Tubulin, Tim23/Tubulin). Source data are provided as a Supplementary Source Data file.

Supplementary Fig. 1g. U2OS and SH-SY5Y cells were treated with 5 μ M UMI-77 for the indicated times and cell lysates were immunoblotted with indicated antibodies. The numbers under the blots represent the gray scale quantification (Tom20/Tubulin, Tim23/Tubulin). Source data are provided as a Supplementary Source Data file.

Supplementary Fig. 1h. U2OS cells were treated with 5 μ M UMI-77 in the presence or absence of MG-132, E64D and NH₄Cl/Leupeptin for 12 h, and cell lysates were immunoblotted with indicated antibodies. The numbers under the blots represent the gray scale quantification (Tom20/Tubulin, Tim23/Tubulin). Source data are provided as a Supplementary Source Data file.

Supplementary Fig. 1i. HEK293T cells were treated with 5 μ M UMI-77 for the indicated times, and cell lysates were immunoblotted with indicated antibodies. The numbers under the blots represent the gray scale quantification (Tim23/Tubulin, p62/Tubulin). Source data are provided as a Supplementary Source Data file.

Supplementary Fig. 5b. HEK293T cells were transfected with pcDNA3.1-MCL-1-WT (wide type) or pcDNA3.1-MCL-1-M (L213A/D218A) plasmid for the indicated times and cell lysates were immunoblotted with indicated antibodies. The numbers under the blots represent the gray scale quantification (Tim23/Tubulin). Source data are provided as a Supplementary Source Data file.

Supplementary Fig. 8a. HEK293T cells were transfected with NIX siRNA for 48 h and treated with 5 μ M UMI-77 for 12 h. Cell lysates were immunoblotted for mitochondrial marker proteins (Tom20, Tim23). The numbers under the blots represent the gray scale quantification (Tom20/Tubulin, Tim23/Tubulin). siNC: scrambled siRNA. Source data are provided as a Supplementary Source Data file.

Supplementary Fig. 8b. As in Supplementary Fig. 8a, except Bnip3 siRNA was used. Source data are provided as a Supplementary Source Data file.

10. The knockdown efficiency has to be shown in experiments using siRNA.

Response:

We thank the reviewer for the comment. We have now added the knockdown efficiency data to the Fig. 2b, 4e and supplementary Fig. 9a, 9c (see below).

mitophagy levels were quantified. One-way ANOVA (data represent mean \pm S.E.M.; n=3. **** p<0.0001, ns, not significant.). siNC: scrambled siRNA. Source data are provided as a Figure 2 Source Data file.

Figure 4e. The mitophagy levels of control cells and *atg5* knockdown cells treated with UMI-77 were analyzed using one-way ANOVA (data represent mean \pm S.E.M.; n=6. **** p<0.0001. ns, not significant). The siRNA knockdown efficiency was shown using western blot. siNC: scrambled siRNA. Source data are provided as a Figure 4 Source Data file.

Supplementary Fig. 9a. Bax knockdown HEK293T-mt-Keima cells were treated with 5 μ M UMI-77 for 12 h. The mitophagy levels were analyzed using t-test (data represent mean \pm S.E.M.; n>6. *** p<0.001, **** p<0.0001.). The siRNA knockdown efficiency was shown using western blot. siNC: scrambled siRNA. Source data are provided as a Supplementary Source Data file.

Supplementary Fig. 9c. Beclin1 knockdown HEK293T-mt-Keima cells were treated with 5 μ M UMI-77 for 12 h. The mitophagy levels were analyzed using t-test (data represent mean \pm S.E.M.; n=11. *** p<0.001, **** p<0.0001). The siRNA knockdown efficiency was shown using western blotting. siNC: scrambled siRNA. Source data are provided as a Supplementary Source Data file.

11. The authors should show the representative images in 2b, 3g and 5a.

Response:

We thank the reviewer for the comment. We have now added the representative images of Fig. 2b, 3g, and 5a to Supplementary Fig. 3a, 3b, and 3d.

Representative image of Figure 2b, 3g, 5a.

Supplementary Fig. 3a. Representative image of Figure 2b. 469nm shows mitochondria and 586nm shows mitochondria in lysosome.

Supplementary Fig. 3b. Representative image of Figure 3g. GFP shows MCL-1 knockdown cells and 586nm shows mitochondria in lysosome.

12. In fig. 6a, the authors should show ratiometric images but not merged images.

Response:

We thank the reviewer for the comment. We have now added the ratiometric images to Fig 6a (see below).

REVIEWERS' COMMENTS

Reviewer #1 (Remarks to the Author):

The authors should be commended for taking the challenge to address most (if not all) the questions in a great way during this COVID-19 pandemic. Thank you. The AAV-injection based MCL-1 overexpression approach which improved memory and reduced pathology in the APP/PS1 mice was very important and novel, supporting the hypothesis the authors proposed.

Some inaccuracies in references should be corrected before acceptance:

- Reference PMID: 31375365 shall be added for the sentence 'In light of this cumulative evidence mitochondrial dysfunction has been suggested as a pivotal event in the initiation of AD, and interventions that bolster mitochondrial health may ameliorate the neurodegenerative pathologies associated with it1, 11.'
- References 11 and 13 are the same.
- For the sentence 'Conversely, mitophagy enhancement reduces A β plaques and Tau tangles in human neuronal cells and ameliorates memory impairment in transgenic mouse models of AD13, 15.' Here, reference#15 should be removed since in this paper it did not link NR to mitophagy but proposed the UPRmt mechanism.
- For the below sentences, there were mistakes in citations. In addition to the reviews which covers both mechanism and/or clinical trial update (PMID: 31577933; PMID: 30922179), original papers which link NAD+ strategy (e.g, via NR supplementation) to mitophagy induction should be cited (PMID: 24813611; PMID: 30742114).
- Reference #PMID: 30742114 should be cited for the statement 'Urolithin A (UA) is a natural, dietary, microflora-derived metabolite, which can induce mitophagy and ameliorate cognitive decline in the APP/PS1 mouse model49, 50.'
- Statement 'However, the mechanism of mitophagy induced by these two drugs is not clear...' should be revised since multiple mechanistic explorations (e.g., to show a PINK-1, Parkin-1, and DCT-1-dependent manner) were reported in #PMID: 30742114 .

From the authors, "Nicotinamide riboside (NR) is a safe and effective activator of neuronal mitophagy47. NR is a precursor of NAD+ and can be metabolized to produce NAD+ in cells. NR reduces A β levels in APP/PS1 mice and has been tested in clinical trials47, 48. Urolithin A (UA) is a natural, dietary, microflora-derived metabolite, which can induce mitophagy and ameliorate cognitive decline in the APP/PS1 mouse model49, 50. However, the mechanism of mitophagy induced by these two drugs is not clear, highlighting the necessity to identify safe, effective, and clear mechanisms for mitophagy inducers that can become effective therapeutic approaches for the treatment of Alzheimer's disease."

Reviewer #2 (Remarks to the Author):

accept

Reviewer #3 (Remarks to the Author):

The authors properly responded to my criticism.

Reviewer #1 (Remarks to the Author):

The authors should be commended for taking the challenge to address most (if not all) the questions in a great way during this COVID-19 pandemic. Thank you. The AAV-injection based MCL-1 overexpression approach which improved memory and reduced pathology in the APP/PS1 mice was very important and novel, supporting the hypothesis the authors proposed.

We appreciate your recognition of the value of our study and help us improve our research results.

Some inaccuracies in references should be corrected before acceptance:

- Reference PMID: 31375365 shall be added for the sentence 'In light of this cumulative evidence mitochondrial dysfunction has been suggested as a pivotal event in the initiation of AD, and interventions that bolster mitochondrial health may ameliorate the neurodegenerative pathologies associated with it¹¹, 11.'

Thank you for your suggestion. We cited the reference.

- References 11 and 13 are the same.

Thank you for pointing out our mistake. We corrected them.

- For the sentence 'Conversely, mitophagy enhancement reduces A β plaques and Tau tangles in human neuronal cells and ameliorates memory impairment in transgenic mouse models of AD¹³, 15.' Here, reference#15 should be removed since in this paper it did not link NR to mitophagy but proposed the UPR^{mt} mechanism.

Thank you for your suggestion. We removed the reference.

- For the below sentences, there were mistakes in citations. In addition to the reviews which covers both mechanism and/or clinical trial update (PMID: 31577933; PMID: 30922179), original papers which link NAD⁺ strategy (e.g, via NR supplementation) to mitophagy induction should be cited (PMID: 24813611; PMID: 30742114).

Thank you for your suggestion. We cited these two original papers (see text below).

"NR is a precursor of NAD⁺ and can be metabolized to produce NAD⁺ in cells which can rescue the function of mitochondria in xeroderma pigmentosum group A⁴⁹. NR reduces A β levels in APP/PS1 mice and has been tested in clinical trials^{48, 50}. Urolithin A (UA) is a natural, dietary, microflora-derived metabolite, which can induce mitophagy and ameliorate cognitive decline in the APP/PS1 mouse model^{11, 51, 52}"

11. Fang, E.F. *et al.* Mitophagy inhibits amyloid-beta and tau pathology and reverses cognitive deficits in models of Alzheimer's disease. *Nat Neurosci* **22**, 401-+ (2019).
49. Fang, E.F. *et al.* Defective mitophagy in XPA via PARP-1 hyperactivation and NAD(+)/SIRT1 reduction. *Cell* **157**, 882-896 (2014).

- Reference #PMID: 30742114 should be cited for the statement ‘Urolithin A (UA) is a natural, dietary, microflora-derived metabolite, which can induce mitophagy and ameliorate cognitive decline in the APP/PS1 mouse model^{49, 50.}’

Thank you for your suggestion. We cited this paper.

- Statement ‘However, the mechanism of mitophagy induced by these two drugs is not clear...’ should be revised since multiple mechanistic explorations (e.g., to show a PINK-1, Parkin-1, and DCT-1-dependent manner) were reported in #PMID: 30742114 .

Thank you for your suggestion. In order not to mislead, we deleted this sentence.

From the authors, “Nicotinamide riboside (NR) is a safe and effective activator of neuronal mitophagy⁴⁷. NR is a precursor of NAD⁺ and can be metabolized to produce NAD⁺ in cells. NR reduces A β levels in APP/PS1 mice and has been tested in clinical trials^{47, 48}. Urolithin A (UA) is a natural, dietary, microflora-derived metabolite, which can induce mitophagy and ameliorate cognitive decline in the APP/PS1 mouse model^{49, 50}. However, the mechanism of mitophagy induced by these two drugs is not clear, highlighting the necessity to identify safe, effective, and clear mechanisms for mitophagy inducers that can become effective therapeutic approaches for the treatment of Alzheimer's disease.”

Reviewer #2 (Remarks to the Author):

accept

We appreciate your recognition of the value of our study

Reviewer #3 (Remarks to the Author):

The authors properly responded to my criticism.

We appreciate your recognition of the value of our study